# Effects of land use and anthropogenic aerosol emissions in the Roman Empire

Anina Gilgen[1], Stiig Wilkenskjeld[2], Jed O. Kaplan[3,4], Thomas Kühn[5,6], and Ulrike Lohmann[1]

[1]ETH Zürich, Institute for Atmospheric and Climate Science, Zurich, Switzerland
[2]Max Planck Institute for Meteorology, Hamburg, Germany
[3]Institute of Geography, University of Augsburg, Augsburg, Germany
[4]Department of Earth Sciences, University of Hong Kong, Hong Kong, China
[5]Atmospheric Research Centre of Eastern Finland, Finnish Meteorological Institute, Kuopio, Finland
[6]Aerosol Physics Research Group, University of Eastern Finland, Kuopio, Finland

**Correspondence:** Anina Gilgen (anina.gilgen@env.ethz.ch) or Ulrike Lohmann (ulrike.lohmann@env.ethz.ch)

**Abstract.** As one of the first transcontinental polities that led to widespread anthropogenic modification of the environment, the influence of the Roman Empire on European climate has been studied for more than 20 years. Recent advances in our understanding of past land use and aerosol-climate interactions make it valuable to revisit the way humans may have affected the climate of the Roman Era. Here we estimate the effect of humans on some climate variables in the Roman Empire at its

apogee, focusing on the impact of anthropogenic land cover as well as aerosol emissions. For this we combined existing land use scenarios with novel estimates (low, medium, high) of aerosol emissions from fuel combustion and burning of agricultural land. Aerosol emissions from agricultural burning were greater than those from fuel consumption, but on the same order of magnitude.

    Using the global aerosol-enabled climate model ECHAM-HAM-SALSA, we conducted simulations with fixed sea-surface

temperatures to gain a first impression about the possible climate impact of anthropogenic land cover and aerosols in the Roman Empire. While land use effects induced a regional warming for one of the reconstructions caused by decreases in turbulent flux, aerosol emissions enhanced the cooling effect of clouds and thus led to a cooling in the Roman Empire. Quantifying the anthropogenic influence on climate is however challenging since our model likely overestimates aerosol-effective radiative forcing and prescribes the sea-surface temperatures.

*Copyright statement.*  TEXT

## 1  Introduction

Humans shaped the European landscape thousands of years before the Industrial Revolution (Kaplan et al., 2016) primarily through deforestation, which was applied for several purposes: less dense forests facilitated hunting, foraging, and mobility (Kaplan et al., 2016), removal of trees was required for most forms of agricultural land use (Klein Goldewijk et al., 2011),

and wood was harvested for fuel or as timber for manufacturing and building purposes (Harris, 2013). The replacement of

forests by other vegetation types influences climate through both biogeochemical and biogeophysical effects (Bathiany et al., 2010). Biogeochemical effects occur primarily through changes in the chemical composition of the atmosphere, e.g. transfers of carbon from land to atmosphere, where it resides as the greenhouse gas $CO_2$. Because atmospheric $CO_2$ is long-lived and well mixed, biogeochemical effects have a global impact on climate (Boysen et al., 2014). In contrast, biogeophysical effects rather act at regional scale. These include changes in physical land surface characteristics, such as roughness length or surface albedo (Claussen et al., 2001). Increases in surface albedo due to deforestation usually lead to a cooling, while changes in energy redistribution (evaporative fraction and turbulent flux) are associated with a warming (Lee et al., 2011). The net effect of biogeophysical effects on temperature is a strong function of radiation and thus latitude (Lee et al., 2011; Li et al., 2016d): in the tropics, changes in energy redistribution usually dominate; as a consequence, deforestation induces a net warming. In high latitudes, changes due to surface albedo are generally more important, and deforesation thus leads to a net cooling.

Smith et al. (2016) modelled the impact of biogeophysical effects during the Holocene and found that anthropogenic land cover change in Europe and East Asia had significant impacts on climate (temperature, precipitation, and near surface wind speed) already a few thousand of years ago. They used the reconstruction of anthropogenic land cover change from Kaplan and Krumhardt (KK10; Kaplan et al., 2009, 2011) for their simulations.

Humans also affect the climate by aerosol emissions. Aerosol particles absorb and scatter radiation, which leads to a radiative forcing due to aerosol-radiation interactions ($RF_{ari}$). Furthermore, aerosols affect clouds by heating atmospheric layers when they absorb a considerable amount of the solar radiation (Koch and Del Genio, 2010). Depending on the cloud type and the position of the aerosol layer relative to the cloud, cloud cover can either decrease or increase (Koch and Del Genio, 2010). Together with $RF_{ari}$, these absorption-related adjustments are called $ERF_{ari}$ (effective radiative forcing due to aerosol-radiation interactions).

Aerosols further impact radiation indirectly by influencing the number of cloud droplets as well as ice crystals. Aerosols that activate into cloud droplets are called cloud condensation nuclei (CCN). If the CCN concentration increases at a fixed cloud liquid water content, then the cloud droplet number concentration (CDNC) generally also increases, which enhances the backscattering of radiation (Twomey, 1974, 1977). This effect is called $RF_{aci}$ (radiative forcing due to aerosol-cloud interactions; Boucher et al., 2013) and generally increases the cooling effect of clouds. An increase in CCN (and thus CDNC) can also have other effects, e.g. affecting the cloud lifetime by decelerating collision-coalescence (Albrecht, 1989). Together with $RF_{aci}$, such adjustments are called $ERF_{aci}$ (effective radiative forcing due to aerosol-cloud interactions; Boucher et al., 2013).

It is not straightforward to disentangle all these different aerosol effects. Here, we define the aerosol radiative effect (ARE) and the cloud radiative effect (CRE) as the differences between radiative fluxes at the top of the atmosphere in the presence and absence of aerosols and clouds, respectively. These quantities can be calculated with a climate model by a double call of the radiation scheme once with and once without aerosols or clouds.

Present-day anthropogenic aerosol emissions are very high compared to pre-industrial emissions. However, when the effect of anthropogenic aerosol emissions on the radiative balance is quantified, it makes a difference whether AD 1850 or AD 1750 is chosen as the reference year (Carslaw et al., 2017). This shows that anthropogenic aerosol emissions probably had an impact

on climate already in AD 1850. But when did anthropogenic aerosol emissions start to change the climate? Is it possible that locally significant changes occurred already thousands of years ago?

In the first century AD, the Roman Empire was at its apogee, with a "surprisingly high standard of living" for antiquity (Temin, 2006). Around the same time, the Han dynasty in China also controlled a large part of the world's population. These two empires had comparable spatial extents and population sizes (Bielenstein, 1986; Scheidel, 2009). Although the global population was approximately 2.6 and 5.5 times smaller in AD 1 than in AD 1700 and AD 1850, respectively (Klein Goldewijk et al., 2017), industrial activites between 100 BC and AD 200 in both Europe and East Asia already left imprints in ice cores and sedimentary records, for example heavy metals and $^{13}$C-enriched methane (Hong et al., 1996; Brännvall et al., 1999; Sapart et al., 2012). In cities and towns during antiquity, smoke emissions were sufficiently severe to darken buildings and to enforce laws against air pollution (Makra, 2015).

Of course, these activities are by no means comparable to the anthropogenic impact on climate under present-day conditions. Nevertheless, we hypothesise that these anthropogenic activities could already have had an influence on climate on a continental scale. The goal of this study is to gain a first impression on whether or not anthropogenic land cover change and aerosol emissions could have influenced climate already in antiquity. We use the global aerosol-climate model ECHAM-HAM-SALSA for this assessment. Since the sea-surface temperatures (SST) are prescribed in all simulations, ocean-atmosphere feedbacks are disabled and the temperature and precipitation responses are dampened. Nevertheless, the results provide valuable information about changes in variables such as surface albedo, turbulent flux, ARE or CRE.

To analyse the influence of land use on climate, we compare a control simulation without land use to simulations including crop and pasture areas representative for AD 100 (Sect. 3.1). Two different reconstructions of anthropogenic land cover were used. Since ECHAM-HAM-SALSA does not calculate a full carbon cycle, we only investigate biogeophysical effects and not the biogeochemical effects. The impact of anthropogenic land cover on secondary organic aerosol (SOA) precursors is simulated.

To assess the impact of anthropogenic aerosol emissions, we first constructed three possible scenarios (called "low", "intermediate", and "high", respectively) of emissions representative for AD 100. We considered emissions from fuel consumption, crop residue burning, and pasture burning. The magnitude of these new aerosol emissions is compared with that of natural fire emissions in Sect. 3.2 and with that of anthropogenic emissions in AD 1850 in Sect. 3.3. Simulations with and without anthropogenic aerosol emissions were compared to quantify their impact on certain climate variables (Sect. 3.4).

## 2  Methods

We conducted several simulations (Sect. 2.8) with an aerosol-climate model (Sect. 2.1). The simulations aim to represent the Roman Empire at its maximum extent around AD 100 (Fig. S1). Our study domain is a box between $10°$ W and $50°$ E, $20°$ N and $60°$ N, which fully encompasses the Roman Empire at that time. We acknowledge that this definition includes some regions that were not part of the Roman Empire, such as the highly populated Northern Germany, but drawing precise boundaries was challenging due to the coarse spatial resolution of our model, and many of the land use and industrial activities present within

the political boundaries of the Roman Empire at this time also occurred outside of it. Since the main focus of our study lies on the anthropogenic influence on climate, and humans near but outside the Roman Empire also carried out agriculture and emitted aerosols, our main conclusions should not be affected by the exact geographical definition. Similarly, some boundary conditions (vegetation, fire emissions) refer to a somewhat earlier period than AD 100 (e.g. AD 1) due to data availability, but concerning the large temporal uncertainties when going so far back in time, we do not consider this to be an issue.

Associated with these experiments, we needed data for i) the boundary conditions (Sect. 2.2), ii) anthropogenic land cover change (Sect. 2.3), iii) natural aerosol emissions (Sect. 2.4), and iv) anthropogenic aerosol emissions (Sect. 2.5, 2.6, 2.7).

## 2.1 Model

To study potential anthropogenic effects of land cover and aerosols, we used the global aerosol-climate model ECHAM6.3-HAM2.3-SALSA2.0. SALSA stands for Sectional Aerosol module for Large Scale Applications (Kokkola et al., 2008; Bergman et al., 2012; Kokkola et al., 2018). The aerosol size distribution is described with 10 size sections. The aerosol species black carbon (BC), organic matter ($OM = 1.4 \cdot OC$, where OC stands for organic carbon), sulfate ($SO_4$), dust, and sea salt are considered. Particles below $r = 25\,nm$ comprise exclusively of OM and/or $SO_4$, whereas larger particles can contain any aerosol species.

The land component of ECHAM-HAM-SALSA is called JSBACH (Jena Scheme for Biosphere–Atmosphere Coupling in Hamburg; Raddatz et al., 2007). In our simulations, heterogeneity in each grid box is represented by geographically varying fractions of 12 different plant functional types. The implementation of anthropogenic land cover change in JSBACH is described in Reick et al. (2013).

## 2.2 Boundary conditions

The greenhouse gas concentrations follow Meinshausen et al. (2017). Both the greenhouse gas concentrations and the orbital parameters were averaged over AD 50-150 (Table S1). To prescribe natural vegetation fractions (Sect. 2.3), SST, and sea ice concentrations (SIC) we used output from a simulation with the Earth System Model MPI-ESM (Bader et al., 2019, in review, Fig. 1), called MPI_no_LCC in the following. The annual cycle of SST and SIC was derived by averaging the MPI_no_LCC output from AD 50 to 150. The MPI-ESM model has the same atmospheric core (ECHAM) and uses the same vegetation model (JSBACH) as ECHAM-HAM-SALSA. MPI_no_LCC ran from 6000 BC to AD 1850 and considered slow forcings, i.e. changes in greenhouse gases ($CO_2$, $CH_4$, $N_2O$) and orbital parameters, but no anthropogenic land cover change. Vegetation was calculated dynamically (Brovkin et al., 2009).

For the simulated sulfur cycle and SOA calculation, climatologies for the following oxidants are needed: $H_2O_2$ (only for sulfur), $O_3$, OH, and $NO_3$. In general, it is uncertain how these oxidants have changed over the past millennia. OH concentrations seem to be relatively stable (Pinto and Khalil, 1991; Lelieveld et al., 2002; Murray et al., 2014). In contrast, ice core measurements suggest that $H_2O_2$ has increased by $> 50\%$ over the last 200 years (Sigg and Neftel, 1991; Anklin and Bales, 1997). Modelling studies suggest that also $NO_3$ and $O_3$ have increased since the pre-industrial (Khan et al., 2015; Murray et al., 2014). For $O_3$, Pinto and Khalil (1991); Crutzen and Brühl (1993) found that the changes are relatively small between the

glacial and pre-industrial; however, the ozone profile shows larger changes between pre-industrial and present-day conditions (Crutzen and Brühl, 1993). In line with this, we expect that differences in oxidant concentrations are larger between AD 1850 and present-day than between AD 100 and AD 1850 due to large anthropogenic emissions of various gases since AD 1850. Therefore, we used climatological monthly mean mixing ratios of oxidants representative for AD 1850 conditions (Fig. 1).

They were derived from simulations with the Community Earth System Model version 2.0 (CESM2.0) Whole Atmosphere Community Climate Model (WACCM[1]).

## 2.3   Vegetation and land cover change

In our simulations with ECHAM-HAM-SALSA, natural vegetation was not dynamic. The coverages of different natural vegetation types representative for AD 100 were taken from MPI_no_LCC (Fig. 1). These natural vegetation fractions are fixed over

time. They are from an earlier year than AD 100 (end of year 10 BC) because vegetation around AD 1 was used to calculate the fire emissions (Sect. 2.4).

For the sensitivity simulations in ECHAM-HAM-SALSA, we used two different reconstructions to estimate anthropogenic land cover fractions around AD 100: the anthropogenic land cover reconstructions from KK11 (an update of KK10; Kaplan et al., 2011, 2012), and the reconstructions from the HYDE database version 3.1 (Klein Goldewijk et al., 2010, 2011, HYDE11

in the following). The empirical model of KK11 assumes that per capita land use declines over time. In contrast, the reconstruction of cropland and pasture from HYDE11 assumes a nearly constant per capita land use hindcasting approach with allocation algorithms that change over time (Klein Goldewijk et al., 2017). As a consequence, the anthropogenic land cover fraction in the past is considerably higher in the estimate by KK11 compared to the estimate of HYDE11.

KK11 provides information about the fraction of a gridbox subject to anthropogenic land use. For our simulations, we

interpolated the values for AD 1 from a $0.5° \times 0.5°$ grid to the Gaussian grid of ECHAM-HAM-SALSA and assumed that the same fraction of natural vegetation is converted to crop and pasture in equal shares. This seems to be a good first order approximation: in the reconstruction of HYDE11, which estimated pasture and crop areas separately, they contribute each roughly 50%, both when averaged over the whole world and when averaged over the study domain. However, there are of course regions (both in HYDE11 and in reality) where either crop or pasture dominated.

HYDE11 provides information about the area of crop and pasture ($km^2$ per gridcell) on a 5' grid. We divided these areas by the maximum land area available per gridcell (also from HYDE11) to get fractions of crop and pasture per land area and interpolated the values to the Gaussian grid of ECHAM-HAM-SALSA.

The anthropogenic land cover fractions were scaled to the fraction of the vegetable part of the model grid. Due to inconsistencies between the data sets and the model setup, including differences in the land sea mask, the actually applied land use

changes are smaller than prescribed in the original data sets. In our model, the total crop areas in the study domain amount to $5.46 \cdot 10^5 \, km^2$ and $13.8 \cdot 10^5 \, km^2$ for HYDE11 and KK11, respectively (i.e. 47% and 21% lower than the original estimate). For pasture, the total areas add up to $4.75 \cdot 10^5 \, km^2$ and $13.8 \cdot 10^5 \, km^2$, respectively (47% and 12% lower than the original

---

[1]   https://svn-ccsm-inputdata.cgd.ucar.edu/trunk/inputdata/atm/cam/tracer_cnst/tracer_cnst_WACCM6_halons_3DmonthlyL70_1850climo295_c180426. nc, downloaded: 27 July 2018

estimate). For comparison, the total land area in the study domain is $163 \cdot 10^5$ km$^2$ in our model. Part of the underestimation in both datasets is due to the binary land sea mask of JSBACH, which can lead to an underestimation along the coast lines: no crop or pasture can grow in gridboxes that are considered to be ocean in JSBACH, even if the anthropogenic land cover fraction from the original estimate would be larger than zero at this location. The underestimation is considerably more pronounced for

KK11 since some areas that are subject to land use in this reconstruction are hardly hospitable to plants in JSBACH. Examples are the Arabian Peninsula, which is and presumably was in reality mainly covered by desert, and parts of North Africa. For instance, the area subject to land use between 35° E and 50° E, 20° N and 35° N (roughly corresponding to the part of the Arabian Peninsula lying within our study domain) amounts to $12.5 \cdot 10^5$ km$^2$ in the original estimate of KK11, but only to $2.28 \cdot 10^5$ km$^2$ in the model.

**2.4  Natural aerosol emissions**

Sea salt, dust, and oceanic dimethylsulfide (DMS) emissions are calculated online as described in Tegen et al. (2019). Tropospheric SO$_2$ emissions from volcanoes are based on Andres and Kasgnoc (1998); Halmer et al. (2002) as described in Stier et al. (2005).

    To estimate SOA formation from biogenic sources, a volatility basis set (VBS) was used (Kühn et al., 2019, in preparation;

Mielonen et al., 2018; Stadtler et al., 2018). The VBS uses three volatility bins following Farina et al. (2010), which are classified as non-volatile (log(C*) = -inf), low-volatile (log(C*) = 0) and semi-volatile (log(C*) = 1), where C* is the equilibrium vapour concentration in $\mathrm{g\,m^{-3}}$ over a flat surface at $298$ K. Gas-to-particle partitioning is computed using a kinetic algorithm after Jacobson (2002). The volatile SOA precursors (monoterpenes and isoprene) are converted into SOA forming species using a simplified one-step oxidation chemistry (Sect. 2.2) and grouped into the volatility bins using pre-defined partitioning

coefficients. The emission strengths of the biogenic SOA precursors were calculated online based on the vegetation in the model (Henrot et al., 2017). As a consequence, all ECHAM-HAM-SALSA simulations that include anthropogenic land use account for the effect of these land cover changes on SOA precursor emissions (Table S2). DMS emissions from terrestrial sources were set to present-day values (Pham et al., 1995); the emissions are very low compared to oceanic DMS emissions, though.

Fires have played an important role in shaping the composition and structure of Mediterranean vegetation communities (Naveh, 1975). To simulate past natural fire emissions, we used a stand-alone version of the carbon and vegetation dynamics sub-model of JSBACH (CBALONE; called CBALANCE in Wilkenskjeld et al., 2014) together with the fire submodel SPITFIRE (Thonicke et al., 2010; Lasslop et al., 2014; Rabin et al., 2017). CBALONE-SPITFIRE needs forcing data related to the vegetation and the atmosphere at a daily time resolution as well as a starting point for the carbon pools. To drive CBALONE-

SPITFIRE we used data from MPI_no_LCC (illustrated in Fig. 1) from which output at a high temporal resolution was saved for a few selected 30-year periods (among them a period around AD 1 and one around AD 1835). The driving data for the AD 100 CBALONE-SPITFIRE simulations thus represents a slightly earlier period (around AD 1). However, we do not expect a significant difference between AD 1 and AD 100 since the forcing in MPI_no_LCC is very similar for the two periods. For simplicity we thus refer to the AD 1 fire emissions as those of AD 100. The 30-year period around AD 1 was repeatedly used

to drive both the spin-up ($\approx 100$ years) and the analysed simulated period of CBALONE-SPITFIRE. The emissions have a daily resolution and show interannual variability.

We also calculated fire emissions around AD 1835 (not shown). These emissions were not used as input for our Roman Empire ECHAM-HAM-SALSA simulations but for comparison with the fire emissions from van Marle et al. (2017, Sect. S9). We again used high time resolution output from an MPI-ESM simulation (hereinafter MPI_LCC) to drive CBALONE-SPITFIRE. The only difference between MPI_no_LCC and MPI_LCC is that the latter considers anthropogenic land cover change.

Next to carbon pools and variables related to the vegetation or the atmosphere, the fire model depends on the following additional inputs: i) lightning; ii) population density; iii) a regionally varying anthropogenic influence factor $a_n$.

To estimate the lightning frequency, we tested the parameterisation by Magi (2015) that is based on convective precipitation fluxes. We found that the such-derived lightning frequency is very similar for AD 100 and AD 1835. However, the simulated lightning frequency differs from the standard lighting frequency used by the MPI-ESM implementation of SPITFIRE (e.g. different spatial pattern; not shown), which is based on the Lightning Imaging Sensor/Optical Transient Detector (LIS/OTD) as described in Lasslop et al. (2014). The difference between the simulated and observation-based lightning frequency is probably associated with the high uncertainties in the parameterised convection scheme. Furthermore, the parameterisation by Magi (2015) works better for convective mass flux through the 0.44 hybrid-sigma pressure level than for convective precipitation, but the former was not available from the MPI-ESM simulations. Thus, we decided to use the observationally derived lightning frequencies by LIS/OTD for both AD 100 and AD 1835 (Fig. 1) in order to not introduce a large bias in natural fire ignitions.

For the CBALONE-SPITFIRE simulations around AD 1835, we averaged the population density from HYDE11 over the years 1830 and 1840. Furthermore, we considered anthropogenic land cover change when calculating the fire emissions; the same changes were used as in MPI_LCC. For AD 100, we used CBALONE-SPITFIRE to calculate natural fire emissions (Fig. 1), assuming no anthropogenic influence. Hence, no anthropogenic land cover change was considered in these CBALONE-SPITIRE simulations. Furthermore, the population density was set to 0. The ECHAM-HAM-SALSA simulations LCC_HYDE_low, LCC_HYDE_int, and LCC_KK_high (Sect. 2.8) account for anthropogenic aerosol emissions, which include agricultural burning. Using the same natural fire emissions in these simulations as in the other simulations (no_human, LCC_HYDE, and LCC_KK) would lead to an overestimation in total aerosol emissions since natural aerosol emissions should not occur where now crop or pasture grows. Therefore, we reduced the natural fire emissions of the simulations LCC_HYDE_low, LCC_HYDE_int, and LCC_KK_high offline to account for regions subject to anthropogenic land use (Table S2; Fig. 1). As a first order approximation, the natural fire emissions calculated with CBALONE-SPITFIRE were multiplied with $(1 - l)$, where $l$ is the fraction of the vegetated area per gridbox that is covered by crop and pasture.

Humans impact fires directly by ignitions and fire fighting as well as indirectly, e.g. by forest management and landscape fragmentation (Arora and Melton, 2018). It is very likely that the relationship between humans and fires has changed in the past centuries and millennia as a consequence of large cultural and political changes. This is neglected in SPITFIRE since $a_n$, which reflects cultural differences, does not change over time. Instead, $a_n$ (and thus the number of anthropogenic ignitions) in SPITFIRE is tuned towards present-day fires where satellite data is available. For estimating the fire emissions in AD 1835, we nevertheless used CBALONE-SPITFIRE with the standard $a_n$, as it was done by the FIREMIP project for calculating fire

emissions from 1750 to today (Rabin et al., 2017). For calculating the (natural) fire emissions around AD 100, $a_n$ is irrelevant because the population density is set to zero.

## 2.5 Aerosol emissions from fuel consumption

Many variables influencing the anthropogenic aerosol emissions are highly uncertain. Therefore, three sets of variables respec-
5 tively leading to low, intermediate and high aerosol emissions were estimated for the Roman Empire based on literature. Figure 3 illustrates how much these scenarios differ.

Anthropogenic emissions associated with both fuel consumption and agricultural burning were likely to have had pronounced regional variations. Despite this variability, we tried to estimate values representative for the whole of our study domain for variables such as fuel consumption or fuel load.

We treated the anthropogenic emissions in the same way as natural fire emissions, except for the emission height. In contrast to the natural fire emissions, for which the simulated emission profile depends on the planetary boundary layer (Veira et al., 2015), we emitted the anthropogenic aerosols at the surface.

For fuel consumption, the aerosol emissions $EM$ of species $i$ [$\text{kg}_{\text{aerosol}}$ m$^{-2}$ s$^{-1}$] were estimated using the following equation:

$$EM_i = cons \cdot popd \cdot EF_i, \tag{1}$$

where $cons$ [$\text{kg}_{\text{dry\_fuel}}$ capita$^{-1}$ s$^{-1}$] is the fuel consumption per capita, $popd$ [capita m$^{-2}$] is the population density, and $EF$ is the aerosol emission factor of species $i$ [$\text{kg}_{\text{aerosol}}$ $\text{kg}_{\text{dry\_fuel}}^{-1}$]. In the following, we derive estimates for these three variables for the different emission scenarios. We assume that the three variables are independent when calculating the emissions.

### 2.5.1 Fuel consumption per capita

In the Roman Empire, people produced aerosol particles by burning several types of fuel for different purposes such as cooking, residential heating, heating bath houses, iron production, glass making, pottery production, or cremation (Malanima, 2013; Veal, 2017; Mietz, 2016; Janssen et al., 2017). For iron production, high temperatures are needed, which were only achieved by burning charcoal (Janssen et al., 2017). For other purposes, also wood or agricultural waste products (e.g. olive pits or dung) were used (Mietz, 2016). The Roman Empire consisted of different regions (e.g. rural versus urban; wetter versus drier climate)
with different fuel consumptions per capita and different fuel strategies. Presently, it is therefore not possible to estimate the fuel consumption with large confidence.

Malanima (2013) estimates that 1-2 kg of wood were consumed per capita per day in the ancient Mediterranean region. His numbers refer to fuel for residential heating and cooking, while he states that industrial contributions were negligible. In present-day developing countries that mainly depend on fuel wood for producing energy (sometimes used as surrogates for
past conditions), typical estimates of domestic fuelwood consumption also range from 1 to 2 kg per capita per day according to Wood and Baldwin (1985). Also the values in Yevich and Logan (2003) are in the same range: for Africa (year 1985), the mean

**Table 1.** The values used to calculate the aerosol emissions from fuel consumption (Equation 1) for the low, the intermediate, and the high scenarios. The total population in the study domain is shown instead of the population density (*popd*, used in the equation) since this is more intuitive.

| Var. | Unit | Low estimate | Intermediate estimate | High estimate |
|------|------|------|------|------|
| *cons* | $kg_{dry\_fuel}$ capita$^{-1}$ day$^{-1}$ | 1.5 | 3 | 5 |
| Population | $10^6$ | 55[a] | 82 | 137 |
| $EF_{combined}$, BC | g $kg_{dry\_fuel}^{-1}$ | 0.23 | 0.37 | 0.59 |
| $EF_{combined}$, OC | g $kg_{dry\_fuel}^{-1}$ | 1.88 | 2.89 | 4.45 |
| $EF_{combined}$, $SO_2$ | g $kg_{dry\_fuel}^{-1}$ | 0.041 | 0.10 | 0.18 |

[a]Klein Goldewijk et al. (2010, 2011)

and median of fuelwood consumption over the countries considered are approximately $1.5\,kg$ per capita per day (assuming 15% moisture content).

Based on the quantitative model from Pompeii, Veal (2017) applied two extreme scenarios for Rome, where the low and the high estimates for fuel consumption are 1 and $2\,t$ per capita per year. This corresponds to 2.7 and $5.4\,kg$ per capita per day, respectively, which is 2.7 times larger than the estimates by Malanima (2013). The fuel estimates by Veal (2017) however refer to fuel (i.e. the sum of wood and charcoal) in contrast to the estimates by Malanima (2013), which refer to wood (either burnt directly or used for charcoal making; he assumes little contribution from the latter). This distinction is important when calculating emissions because several kilogrammes of wood are needed to make one kilogramme of charcoal, making the difference between the two estimates even larger. While Veal's model derived for Pompeii might give reasonable results for Rome, it is likely not applicable to all parts of the Roman Empire, e.g. in the countryside. Nevertheless, it shows that the fuel consumption in the Roman Empire might have been substantially larger than suggested by Malanima (2013), since the estimates of Veal (2017) account for all types of fuel consumption, including e.g. baths and industrial activites. Recently, Janssen et al. (2017) calculated the wood consumption for the city Sagalassos (2500-3500 inhabitants) to range between 0.6 and $0.8\,kg$ per capita per day for local pottery production and 1.3-3.4 kg per capita per day for heating the bath (oven dry wood). Although Sagalassos might differ from other places, this indicates that the neglect of non-residential sources by Malanima (2013) might not be justified, at least in some regions. Based on these different studies, we use 1.5, 3, and $5\,kg$ of fuel (expressed as wood, wood used for charcoal making, or agricultural waste on a dry fuel mass basis) per capita per day for the low, the intermediate, and the high emission scenarios, respectively (Table 1).

Although more fuel for heating was generally consumed where and when it was cold (Malanima, 2006; Warde, 2006), we assume a constant fuel consumption over the year and over latitudes since we do not differentiate between heating, cooking, iron production, and other burning activities in our calculation. Calculating emissions for individual sectors can be very challenging; as an example, recycling of glass was common (Stern, 1999; Freestone, 2015), which needs to be considered when estimating fuel consumption associated with glass making. Given the large uncertainties about the relative importance of residential heating to total fuel consumption, this seems justified as a first order approximation.

### 2.5.2  Population size

We base the population density of our study on HYDE11 for the year AD 100 (for sources see Klein Goldewijk et al., 2010, 2011). We divided the population counts from HYDE11 (5' grid) by the gridbox area before interpolating the such derived population densities to the Gaussian grid of ECHAM-HAM-SALSA. Using this approach, the HYDE11 population size between 10° E and 50° E, 20° N and 60° N is around 55 million people (the number of people living within the political boundaries of the Roman Empire would be somewhat smaller). However, the population size of the Roman Empire is still debatable since there is disagreement what the census tallies represent. The number of HYDE11 lies in the range of the so-called "low count" scenario (Scheidel, 2008), but there are also proponents of a "middle count" and a "high count" hypothesis (Hin, 2013; Scheidel, 2008). With the "middle count" approach, Hin (2013) arrived at 6.7 million (range between 5 and 10 millions) free citizens in Italy compared to approximately 4 million with the "low count" for 28 BCE. Assuming a constant ratio between free citizens and all people and a similar scaling factor outside Italy, the population in the Roman Empire would be a factor of 1.25-2.5 higher in the "middle count" approach compared to the "low count" approach. The "high count" would result in a roughly 3 times larger population size, i.e. more than 100 million people (maybe up to 160 million) living in the whole Empire if we assume that the population densities in other parts of the Empire were similar to Italy (Scheidel, 2008, 2009).

Based on these different literature values, we used the estimate from HYDE11 for our low emission scenario. For the intermediate and the high emission scenarios, we decided to multiply the population densities of the HYDE database with factors of 1.5 and 2.5, respectively.

### 2.5.3  Aerosol emission factors

Last but not least, we needed to estimate aerosol emission factors from burning biofuel. We compiled an overview of BC, OC, and $SO_2$ emission measurements from different studies (Sect. S10, Table S17). Composite estimates were not considered. We treated BC and elemental carbon (EC) to be the same. We grouped the measurements according to the fuel type, i.e. wood (key 1 in Table S17), agricultural waste (key 2), charcoal burning (key 5), and charcoal production (key 6). Woody agricultural waste was counted as wood. We neglected coal as a fuel type although it was widespread in Roman Britain (Smith, 1997). This is justified since the centres of the classical civilisations (especially the Mediterranean region) were not rich in coal (Malanima, 2013). A few measurements refer to PM10 (i.e. aerosol/particulate matter with aerodynamic diameters $< 10\,\mu m$) or PM4 instead of PM2.5, but the difference is usually small (a few percent between PM10 and PM2.5 in Turn et al., 1997). Similarly, we neglected the difference between $SO_2$ and $SO_x$ since the latter is dominated by $SO_2$.

For the different sectors (i.e. wood, agricultural waste, charcoal burning, and charcoal production), we calculated a lower, an intermediate, and an upper estimate for $EF$ after the following procedure:

– For all measurements, the mean, the standard deviation, and the number of samples $N$ were collected.

– If $N$ was larger than 1, but the standard deviation was not given and could not be calculated (e.g. from plots, confidence intervals, or data in the Supplementary Material), we estimated it by assuming a coefficient of variance of 50% for OC

and BC and of 80% for $SO_2$. We assessed these coefficients of variance from all other observations providing standard deviation and mean. The samples for which the standard deviation was estimated are marked in Table S17.

– If the sample size for measurements was given as a range (e.g. "3 or 4"), we always took the lower number as $N$. When $N$ was larger than 1 but not given, we assumed $N = 2$.

– Following Bond et al. (2004), we assumed that $EF$s follow a log-normal distribution. From the sample mean $m$ and standard deviation $s$, the mean $\mu$ and the standard deviation $\sigma$ of the log-normal distribution were calculated:

$$\mu = \ln\left(\frac{m^2}{\sqrt{s^2 + m^2}}\right) \tag{2}$$

$$\sigma = \sqrt{\ln\left(\frac{s^2}{m^2} + 1\right)} \tag{3}$$

– For each emission sector, we randomly drew samples from the log-normal distributions with the calculated $\mu$ and $\sigma$.

– We used three different methods how to weight the different samples: i) every measurement (in Table S17) had the same weight; ii) the measurements were weighted with $N$; iii) every study had the same weight (differentiated by horizontal lines in Table S17). The three weighting methods can result in very different estimates, e.g. for OC emission factors from wood combustion (medians of $2.4\,\mathrm{g\,kg^{-1}}$, $0.90\,\mathrm{g\,kg^{-1}}$, and $3.0\,\mathrm{g\,kg^{-1}}$, respectively).

– From the randomly generated samples, we calculated the median and the lower and upper quartile for each weighting method (the median is somewhat smaller than the expected value; Bond et al., 2004). The medians of the three weighting methods were then averaged, and the same was done for the quartiles.

We considered the median to be the intermediate estimate for $EF$ and the quartiles to be the lower and the upper estimates. We did not choose more extreme percentiles for the lower and the upper estimates because the large variability in the measurements of $EF$ reflects the high variability in burning conditions (e.g. smoldering versus flaming), fuels, combustion devices, and measurement devices. We explicitly wanted to consider these different conditions and not only sample from one (or a few) measurements conducted under specific conditions. In general, more measurements for BC and OC than for $SO_2$ are available; however, since $SO_2$ emissions from biomass burning are small, we do not consider this to be an issue. In Sect. S3, the estimated $EF$s for the different sectors as well as the weighting of these different sectors are discussed. As described there, we estimate that 20% of the fuel consisted of agricultural waste, 40% of charcoal (in terms of wood that needs to be converted to charcoal), and 40% of wood. The combined aerosol emission factors were thus calculated as:

$$EF_{\mathrm{combined}} = 0.2 \cdot EF_{\mathrm{agr}} + 0.4 \cdot EF_{\mathrm{ch_w}} + 0.4 \cdot EF_{\mathrm{wood}}, \tag{4}$$

For the intermediate scenario, we inserted the medians for $EF_{\mathrm{agr}}$, $EF_{\mathrm{ch_w}}$, and $EF_{\mathrm{wood}}$. For the low and the high estimates, the lower and the upper quartiles were used, respectively. The values for $EF_{\mathrm{combined}}$ can be found in Table 1.

## 2.6 Aerosol emissions from crop residue burning

In the Greco-Roman world, fire was widely employed to fertilise fields, to create or regenerate pastures, to control pests, or to hunt (Ascoli and Bovio, 2013). In all simulations where we included the impact of anthropogenic aerosols, we considered the impact of humans on fires by (i) reducing the (natural) fire emissions calculated from CBALONE-SPITFIRE to account for crop and pasture areas (Sect. 2.4) and (ii) estimating fire emissions from pasture and crop residue burning. In the following, we will first estimate aerosol emissions from crop residue burning before deriving estimates for pasture burning.

Next to fallow, crop rotation, and green manuring (i.e. plowing legumes in the soil; White, 1970), burning crop residues on the field was one method to increase the fertility of soils in Roman agriculture (Spurr, 1986). Since crop residues are burnt after harvest, emissions from open crop residue burning have a strong seasonal cycle (as for example shown for present-day China by Li et al., 2016b; Zhang et al., 2016). Today, harvest of cereal in the Mediterranean region takes place approximately from the beginning of May to the end of August[2], which is in accordance with summer being the time of harvest in Roman Italy (Spurr, 1986). We thus spread the fire emissions from crop residue burning over these four months; we neglected that a part of the harvest might have been burnt after August due to drying in the field. Since fires can get out of control at very high temperatures and are unlikely to be ignited at very low temperatures (Pfeiffer et al., 2013), we assume that crop burning cannot occur when the monthly surface temperature averaged over 20 years is below $0\,°C$ or above $30\,°C$. The emitted mass for these months without burning was shifted to the other months if there were any. The regions where temperature exceeded these thresholds were calculated offline using the surface temperature simulated with ECHAM-HAM-SALSA (without human impact; climatological analysis).

The aerosol emission fluxes $[\mathrm{kg\,m^{-2}\,s^{-1}}]$ were calculated with the following equation (adapted from Webb et al., 2013):

$$EM_{\mathrm{crop}} = Fr_{\mathrm{crop}} \cdot Y \cdot d \cdot s \cdot p_b \cdot Fr_{\mathrm{crop\_burnt}} \cdot C_f \cdot EF_{\mathrm{crop}}, \tag{5}$$

where $Fr_{\mathrm{crop}}$ is the fraction of the total gridbox covered by crop, $Y$ is the crop harvest fresh weight $[\mathrm{kg_{crop}\,m^{-2}}]$, $d$ is the dry matter content of the yield [-], $s$ is the ratio between the residue mass and the crop yield mass $[\mathrm{kg_{residue}\,kg_{crop}^{-1}}]$, $p_b$ [-] is the proportion of residue which is burnt (and not used for other purposes such as fuel consumption), $Fr_{\mathrm{crop\_burnt}}$ is the fraction of crop that is burnt per time $[\mathrm{s^{-1}}]$, $C_f$ is the combustion factor [-], i.e. the proportion of available fuel that is actually burnt, and $EF_{\mathrm{crop}}$ is the emission factor of crop residue $[\mathrm{kg_{aerosol}\,kg_{residue}^{-1}}]$. We consider low, intermediate, and high estimates for $Fr_{\mathrm{crop\_burnt}}$ and $EF_{\mathrm{crop}}$ because these variables have high uncertainties. Since $Fr_{\mathrm{crop}}$, $Y$, and $p_b$ depend on the population size (below), they also have different values for the different scenarios. All values are summarised in Table 2.

### 2.6.1 Crop yield $Y$, fraction of crop $Fr_{\mathrm{crop}}$, and proportion of residue burnt $p_b$

Cereals were the nutritional basis in the Roman Empire (Witcher, 2016) and typical yields were recorded by the agronomists of Classical Antiquity. Based on ancient sources and the work of Goodchild (2007) and Hopkins (2017), we estimated that yields representative for the whole Roman Empire are in the range between $500\,\mathrm{kg\,ha^{-1}}$ and $1000\,\mathrm{kg\,ha^{-1}}$ (Sect. S4).

[2]http://www.claas.de/faszination-claas/themen/erntekalender-weltweit, last access: 23 August 2018

**Table 2.** The values needed to calculate the aerosol emissions from crop residue burning (Equation 5) for the low, the intermediate, and the high scenarios. The total area of crop in the study domain (Area$_{crop}$) is shown instead of the fraction per gridbox ($Fr_{crop}$). The bold values for $Y$ show which values were taken for the calculation, and the values in brackets represent the yields considering fallow.

| Var. | Unit | Low estimate | Intermediate estimate | High estimate |
|---|---|---|---|---|
| $Y$ (HYDE11) | kg ha$^{-1}$ | **440** | **660** | 1100 |
| | | (660-880) | (990-1320) | (1650-2200) |
| $Y$ (KK11) | kg ha$^{-1}$ | 170 | 260 | **430** |
| | | (255-340) | (390-520) | (645-860) |
| Area$_{crop}$ | $10^5$ km$^2$ | 5.46 | 5.46 | 13.9 |
| $Fr_{crop\_burnt}$ | % yr$^{-1}$ | 20 | 40 | 80 |
| $s$ | - | 1.9 | 1.9 | 1.9 |
| $d$ | - | 0.85 | 0.85 | 0.85 |
| $p_b$ | - | 0.87 | 0.74 | 0.56 |
| $C_f$ | - | 0.9 | 0.9 | 0.9 |
| $EF_{crop}$, BC | g kg$^{-1}_{dry\_fuel}$ | 0.28 | 0.52 | 0.77 |
| $EF_{crop}$, OC | g kg$^{-1}_{dry\_fuel}$ | 1.31 | 2.36 | 4.56 |
| $EF_{crop}$, SO$_2$ | g kg$^{-1}_{dry\_fuel}$ | 0.095 | 0.29 | 0.53 |

It is fair to assume that the amount of crop produced scaled with the population size. Total crop production depends both on the crop area and the crop yield. The variables $Fr_{crop}$ and $Y$ can therefore not be estimated independently. Following Kessler and Temin (2007, estimate for Rome), we assumed that around $0.8$ kg of wheat (which was the most important nutrient) was consumed per person per day. We roughly estimated that the wheat production was around 50% higher than the wheat

consumption (resulting in $1.2$ kg of crop yield per person per day) to consider that a part of the produced crop yield was lost through, e.g. transport or insect damage (Spurr, 1986), used for fodder (Spurr, 1986), or needed for seeding (Hopkins, 1980).

For the low emissions scenario where the population size originates from HYDE11, this wheat production results in a yield of $440$ kg ha$^{-1}$ with the crop area based on HYDE11. For consistency, we use the simulated crop areas from ECHAM-HAM-SALSA (which are somewhat smaller; Sect. 2.3) for this calculation instead of the original input data. This yield is a bit lower

than the estimates of crop production mentioned above (500-1000 kg ha$^{-1}$). However, considering that part of the crop area was used for other purposes than planting cereals or legumes (e.g. vineyards, which were not burnt) and that the crop area estimates from HYDE11 include areas that can lie fallow, the number seems reasonable: if we assume that fallow took place every second or third year (Goodchild, 2007), "actual" crop yields increase from $440$ kg ha$^{-1}$ to $\approx$ 660-880 kg ha$^{-1}$. For the intermediate scenario where population size is larger than in the low emission scenario, the HYDE database results in high

but still reasonable estimates of crop yield ($660$ kg ha$^{-1}$; $\approx$ 990-1320 kg ha$^{-1}$ if fallow is considered). The high population density in the high emission scenario would require an unrealistically high crop yield when combined with the crop area of HYDE11 ($1100$ kg ha$^{-1}$; 1650-2200 kg ha$^{-1}$ with fallow). When using the KK11 land use reconstruction, which has a much

larger crop area, the needed crop yield would be $430\,\mathrm{kg\,ha^{-1}}$ ($\approx$ 645-860 $\mathrm{kg\,ha^{-1}}$), which is regarded as realistic. Therefore, the HYDE11 land use is chosen for the low and the intermediate scenarios while KK11 is chosen for the high scenario.

For the yields, we took the values mentioned above (Table 2), which are calculated based on the crop area and the population size, for our calculations. Since our approach assumes a constant wheat consumption per person per day across all scenarios, we assure that crop yield provides in all cases the necessary food to feed the entire population.

In Sect. 2.5, we assumed that 20% of the fuel consisted of agricultural waste. Therefore, we should account for the fact that the higher the assumed population in our scenarios, the more crop residue is taken from the field. On the one hand, part of this agricultural waste used as fuel consisted not of cereal crop residue but e.g. of dung or olive pits, which speaks for a larger $p_b$. On the other hand, also other purposes of crop residue than fuel depend directly on the population density, e.g. the residue mass that was used for construction purposes or for filling mattresses (Spurr, 1986), which speaks for a lower $p_b$. Here we simply assume that the crop residue taken from the field per person is equal to 20% of the consumed fuel mass, and thus arrive at $p_b = 0.87$, $p_b = 0.74$, and $p_b = 0.56$ for the low, the intermediate, and the high scenarios, respectively.

### 2.6.2 Fraction of crop burnt $Fr_{\mathrm{crop\_burnt}}$

Part of the remaining residue on the field was burnt. We assume that the fraction of residue burnt in the field did not depend on the remaining residue mass per area but rather on cultural practices. There are several options for what happened after harvest with the remaining residue on the field: when chaff was short, straw could be used as livestock feed (Spurr, 1986) or the residues left could be directly grazed by ruminants (Spurr, 1986), as it happens also nowadays e.g. in Mediterranean North Africa (Yevich and Logan, 2003). However, Spurr (1986) states that animals rarely eat stubble but rather the edible weeds and grass which grow among it. Farmers that had no further use for the stubble left after harvesting, burnt it (Spurr, 1986). In ancient sources, burning was mentioned as method to destroy weeds or to fertilise the soil (Spurr, 1986). Spurr (1986) assumed that especially small farmers who could not store straw and had few animals to feed would burn the stubble.

Nowadays, crop residue burning is no longer widespread in developed countries, and many countries in western Europe even forbid open field burning (Yevich and Logan, 2003). To have a rough indication for how much of the remaining residue might have been burnt in the Roman Empire, we used present-day estimates from developing countries as an indication (including countries that cultivate rice despite the fact that rice was not grown in the Roman Empire). Yevich and Logan (2003) found that in 1985 of the available residue, 1%, 23%, and 38% was burnt in the fields in China, in India, and in the rest of Asia, respectively. A more recent estimate for crop residue burning in China arrives at much higher fractions than the study by Yevich and Logan (2003): based on satellite data, Li et al. (2016a) find that 23% of the field area is burnt. A large part of the discrepancy can be explained by the different year: Yevich and Logan (2003) derived estimates for 1985, whereas Li et al. (2016a) analysed the year 2012. Field surveys from China show that the proportion of residue that is burnt is larger when the crops are harvested by a combine harvester compared to manual harvesting (Yang et al., 2008). In line with this, Li et al. (2016b) estimated that the fractions of crop residues burnt in fields increased from 5% in 1990 to 23% in 2013. Another reason why crop residue burning has increased might be that the use of biofuels has decreased in rural China (Yang et al., 2008). We do not expect that mechanisation generally leads to enhanced crop residue burning; modern-day technology can also be

used to prepare fields for the next crop planting after harvest (Pfeiffer et al., 2013), which makes the burning of crop residue unnecessary.

In the Philippines and Indonesia, up to 65% and 73% of the residue, respectively, was burnt in fields in 1985 (Yevich and Logan, 2003). This is similar to the satellite-derived fraction of rice fields under burning for the Punjab region in India in 2015 (66%, PRSC, 2015); as reasons for the large burning in the Punjab region, highly mechanised farming, poor storage facility for the straw, and lack of market demand for further use are mentioned among others (PRSC, 2015). A survey in Bangladesh showed that only 3% of the farmers burn the total residue but that 37% burn the lower part of it (Haider, 2011); the surveyed farmers manually gathered the residue from the field.

By the use of $p_b$, we have already accounted for the part of residue used as biofuel. If we remove this part of the residue from the estimates above, then the fractions of burning in the field would increase. Based on all the studies that we have mentioned, we estimate that 40% of the crop area is annually burnt for the simulation LCC_HYDE_int ($Fr_{\text{crop\_burnt}} = 0.4\,\text{yr}^{-1}$). For the low and the high scenarios, we changed this fraction by a factor of 2 and arrive at 20% and 80%, respectively. The 80% are reached under present-day conditions in the countries with most crop residue burning (e.g. Indonesia). Pfeiffer et al. (2013) used a value of 20% in their pre-industrial fire model (LPJ-LMfire v1.0) and consider this to be a conservative estimate. Fallow as well as the part of the residue taken from the field for fuel combustion are already accounted for in the derivation of $Y$ and $p_b$.

### 2.6.3 Other variables in Equation 5

Modern values of the harvest index (= the ratio of grain yield to the total plant mass) for wheat range between 0.4 and 0.6 (Hay, 1995). The harvest index was typically lower (around 0.3) at the end of the nineteenth century (Hay, 1995; Sinclair, 1998), but the sometimes exceptionally high yields in ancient times could indicate that the harvest index might have been higher at this time (Sinclair, 1998). For all simulations, we used a harvest index of 0.35, which corresponds to $s = 1.9$. Following Webb et al. (2013), we chose $d = 0.85$ and $C_f = 0.9$. The emission factors for burning of crop residues on the field were estimated using the same method as described in Sect. 2.5.3 using values from Table S17 (key 3).

### 2.7 Aerosol emissions from pasture burning

According to the agronomist Columella (who lived in the first century AD), pasture from long fallow was burnt in late summer to achieve more tender growth (Spurr, 1986). Based on Aalde et al. (2006, Equation 2.27), we calculated the aerosol emission fluxes [$\text{kg}\,\text{m}^{-2}\,\text{s}^{-1}$] with:

$$EM_{\text{pasture}} = Fr_{\text{pasture}} \cdot Fr_{\text{pasture\_burnt}} \cdot F \cdot EF_{\text{pasture}}, \tag{6}$$

where $Fr_{\text{pasture}}$ is the fraction of the total gridbox covered by pasture, $Fr_{\text{pasture\_burnt}}$ is the fraction of pasture that is burnt per time [$\text{s}^{-1}$], $F$ stands for fuel biomass consumption [$\text{kg}_{\text{dry\_matter}}\,\text{m}^{-2}$; i.e. the amount of fuel that is actually burnt], and $EF_{\text{pasture}}$ is the emission factor of pasture burning [$\text{kg}_{\text{aerosol}}\,\text{kg}_{\text{dry\_matter}}^{-1}$]. In accordance with the crop residue burning

**Table 3.** The values used to calculate the aerosol emissions from pasture burning (Equation 6) for the low, the intermediate, and the high scenarios. Note that the total area of pasture in the study domain ($\mathrm{Area_{pasture}}$) is shown instead of the fraction per gridbox ($Fr_{\mathrm{pasture}}$).

| Var. | Unit | Low estimate | Intermediate estimate | High estimate |
|---|---|---|---|---|
| $\mathrm{Area_{pasture}}$ | $10^5\ \mathrm{km}^2$ | 4.75 | 4.75 | 13.9 |
| $Fr_{\mathrm{pasture\_burnt}}$ | $\%\,\mathrm{yr}^{-1}$ | 15 | 30 | 60 |
| $F$ | $\mathrm{kg\,m}^{-2}$ | 0.35 | 0.35 | 0.35 |
| $EF_{\mathrm{pasture}}$, BC | $\mathrm{g\,kg^{-1}_{dry\_fuel}}$ | 0.48 | 0.62 | 0.81 |
| $EF_{\mathrm{pasture}}$, OC | $\mathrm{g\,kg^{-1}_{dry\_fuel}}$ | 4.85 | 6.64 | 7.68 |
| $EF_{\mathrm{pasture}}$, $SO_2$ | $\mathrm{g\,kg^{-1}_{dry\_fuel}}$ | 0.31 | 0.41 | 0.53 |

emissions, we used the land cover reconstructions from HYDE11 for the low and the intermediate scenarios and the land cover estimates from KK11 for the high emission scenario. With this approach, the pasture area per person lies in a range between $0.58\,\mathrm{ha}$ and $1.0\,\mathrm{ha}$ for the different emission scenarios, which is similar to the values ($0.56\,\mathrm{ha}$ and $1.05\,\mathrm{ha}$, respectively) mentioned in Klein Goldewijk et al. (2011, 2017) but somewhat lower than the number derived in a case study for Greece in the Roman period ($1.75\,\mathrm{ha}$; Weiberg et al., 2019). We considered low, intermediate, and high estimates for $Fr_{\mathrm{pasture\_burnt}}$ and the emission factors because these variables have large uncertainties.

### 2.7.1 Fuel biomass consumption $F$

The increase in European grassland productivity over the last decades has been small compared to crop (Smit et al., 2008). The spatial variability of grassland productivity is quite large within Europe, ranging from $\approx 0.15\,\mathrm{kg\,m}^{-2}\,\mathrm{yr}^{-1}$ in the Mediterranean region up to $0.65\,\mathrm{kg\,m}^{-2}\,\mathrm{yr}^{-1}$ in the Atlantic zones; the median over the different climate zones of Europe is $0.33\,\mathrm{kg\,m}^{-2}\,\mathrm{yr}^{-1}$ (Smit et al., 2008). As a consequence, we expect that also the fuel load and the fuel biomass consumption show spatial variability. In contrast to fuel biomass consumption, fuel load refers to fuel that is available (but not necessarily burnt) and often refers to aboveground biomass only. Nevertheless, values for fuel biomass consumption and fuel load are quite comparable because grass easily burns and mainly aboveground biomass is consumed during pasture burning.

Values of fuel loads ($\mathrm{kg_{dry\_matter}}$ per area) frequently lie in the range between 0.3 and $0.5\,\mathrm{kg\,m}^{-2}$ for pastures and grasslands: for Kentucky Bluegrass, 0.26, 0.36, and $0.64\,\mathrm{kg\,m}^{-2}$ were measured in Idaho (Holder et al., 2017); values for grasslands in Australia are around $0.32\,\mathrm{kg\,m}^{-2}$ and $0.46\,\mathrm{kg\,m}^{-2}$ (Cruz et al., 2016, 2017); in South Africa, total aboveground fuel loads in savanna parklands range from 0.22 to $0.55\,\mathrm{kg\,m}^{-2}$ (except for a site subjected to 38 years of fire exclusion; Shea et al., 1996), whereas the fuel loads of standing herbaceous material range from 0.30 to $0.41\,\mathrm{kg\,m}^{-2}$ (Smith et al., 2013); a total fuel load of $0.49\,\mathrm{kg\,m}^{-2}$ for Mediterranean grasslands in Greece was measured (Dimitrakopoulos, 2002).

Fuel biomass consumptions for savannas are comparable: $0.26\,\mathrm{kg\,m}^{-2}$ for savanna woodlands for early dry season burns, $0.46\,\mathrm{kg\,m}^{-2}$ for savanna woodlands for mid/late dry season burns, $0.21\,\mathrm{kg\,m}^{-2}$ for savanna grasslands for early dry season burns, and $1.0\,\mathrm{kg\,m}^{-2}$ for savanna grasslands for mid/late dry season burns ($0.54\,\mathrm{kg\,m}^{-2}$ when excluding one high value for tropical pasture; Aalde et al., 2006).

Based on these studies, we estimate that a reasonable average number for fuel biomass consumption is $F = 0.35 \, \mathrm{kg_{dry\_matter}} \, \mathrm{m}^{-2}$.

### 2.7.2 Fraction of pasture burnt $Fr_{\mathrm{pasture\_burnt}}$

In the Mediterranean region, pasture burning has continued throughout history to the present day and is one component that has shaped the diversity of Mediterranean landscapes as we know them (Montiel and Kraus, 2010). In temperate and boreal Europe, the burning of grasslands for pasture was common on lands which were too poor in nutrients for agriculture, e.g. heathlands (Montiel and Kraus, 2010).

In general, the abundance of prescribed burning depends on the accumulation of biomass: the higher the accumulation, the shorter the fire interval is. As a consequence, the fire interval depends on rainfall and grazing pressure (Weir et al., 2013; Frost and Robertson, 1987), thus showing pronounced regional variability. In the following we summarise guidelines for the rate of prescribed burning from different regions around the world.

For phryganic rangelands in Greece, it is recommended to set fire every 3 to 4 years, which allows to have a good herbage production and at the same time to suppress undesirable dwarf shrub (Papanastasis, 1980). In South Africa, Oluwole et al. (2008) found that the recovery period should be 3 years for optimum productivity in the absence of grazing. In line with this, the Burning Guidelines of South Africa do not recommend to burn pasture every year (what some farmers do), but every 2-5 years in mesic and coastal grasslands and only when it is needed in dry highveld grasslands (SANBI, 2014). Smith et al. (2013) found that grass richness, evenness, and diversity was high for sites with high rainfall when frequent burning was applied in the dry season (1- to 3-year return intervals), whereas Little et al. (2015) conclude that annual burning combined with intensive grazing has a detrimental effect on plant species diversity and structure. In Australia, single fires caused a short-term reduction of yield and cover of pastures in the following year, but fast recovery occurred for most burning regimes. However, perennial grasses were reduced at the expense of annual grasses, which is why burning every 5-6 and 4-6 years for arid short grass and ribbon blue grass, respectively, are recommended (Dyer, 2011). This is in agreement with the findings of Norman (1963, 1969) for native pasture on Tippera clay loam in the Katherine region. For North America, the recommended fire-return-interval of prescribed patch burning is 3 years in areas with rainfall above $\approx 760 \, \mathrm{mm}$ per year and 4 years in drier regions (Weir et al., 2013).

One could argue that farmers in the past did not necessarily follow these present-day guidelines. However, traditional knowledge of prescribed burning has been lost in many European areas (Montiel and Kraus, 2010). Guidelines thus partially re-establish knowledge that our ancestors had. On the one hand, if burning every year reduces the productivity of many grasslands, we think that it is unlikely that ancient farmers burnt their fields annually. On the other hand, a too long period without burning is also unlikely since this e.g. allows the growth of unwanted species and can have adverse effects on the ecosystem (SANBI, 2014; Papanastasis, 1980). According to the summarised literature, burning every $\approx 3$ years seems to be a reasonable intermediate estimate. Therefore, we assumed that $30\%$ of the pasture area is burnt per year for the intermediate scenario. For the low and the high emission scenarios, we changed this fraction by a factor of 2 and thus arrive at $15\%$ and $60\%$, respectively. We assumed that the aerosols from pasture burning were emitted throughout the year, but – like for crop

residue – that no pasture was burnt in months which are very cold or hot (monthly average temperatures below $0\,^\circ\text{C}$ or above $30\,^\circ\text{C}$).

### 2.7.3 Emission factor $EF_{\text{pasture}}$

Using the method described in Sect. 2.5.3 and the measurements from Table S17 (key 4), the emission factors in Table 3 were derived. We again used the lower quartiles for the low, the medians for intermediate, and the upper quartiles for high emission scenarios.

### 2.8 Simulations

We conducted six time-slice experiments representative for the period around AD 100 (Table 4): no_human, LCC_HYDE, LCC_KK, LCC_HYDE_low, LCC_HYDE_int, and LCC_KK_high. The simulation no_human does not account for anthropogenic impacts (except for potential influences on greenhouse gas concentrations at this time). LCC_HYDE and LCC_KK consider only the effects of anthropogenic land cover change (Sect. 2.3) on climate. The simulations LCC_HYDE_low, LCC_HYDE_int, and LCC_KK_high consider, in addition to anthropogenic land cover change, anthropogenic aerosol emissions due to fuel consumption, crop residue burning, and pasture burning. While the anthropogenic land cover reconstructions provide global values and are thus applied on the global scale, we only calculated anthropogenic aerosol emissions for the Roman Empire and not for other parts of the world, i.e. no anthropogenic aerosol emissions occur outside of our study domain.

The simulations have a spatial resolution of approximately $1.875° \times 1.875°$ and 47 vertical levels (T63L47). After the first three months of spin-up, natural vegetation was replaced by the area of anthropogenic land cover for AD 100 within one model year; after that the vegetation fractions were kept constant. One year of model spin-up was added after the transition to anthropogenic land cover was completed, resulting in a total spin-up of 2 years and three months. The spin-up is relatively short since we use neither an interactive ocean nor dynamic vegetation. The simulations (excluding spin-up) are 20 years long and should be representative for approximately AD 100, i.e. the time of interest.

In the standard ECHAM-HAM-SALSA version, the cloud droplet number concentration in a cloud cannot be lower than $40\,\text{cm}^{-3}$ (or optionally $10\,\text{cm}^{-3}$; Tegen et al., 2019). For all simulations of this study, this minimum CDNC was lowered from $40\,\text{cm}^{-3}$ to $1\,\text{cm}^{-3}$ (and the model was retuned) because the aerosol concentrations – and therefore most likely CDNCs – were considerably lower around AD 100 than today.

## 3 Results

### 3.1 Climate impact of anthropogenic land cover change

The simulations LCC_HYDE and LCC_KK were compared with the simulation no_human to quantify the impact of anthropogenic land cover change. Tables S8, S9 show seasonal and annual mean changes in some climate variables, including all that are discussed below.

**Table 4.** Overview of the different simulations. Note that the anthropogenic land cover change is applied in ECHAM-HAM-SALSA, but not in CBALONE-SPITFIRE, which was used to calculate aerosol emissions from natural fires exclusively.

| Simulation | Anthropogenic land cover | Anthropogenic aerosol emissions |
|---|---|---|
| no_human | - | - |
| LCC_HYDE | HYDE11[a] | - |
| LCC_KK | KK11[b] | - |
| LCC_HYDE_low | HYDE11[a] | low estimate[c] |
| LCC_HYDE_int | HYDE11[a] | intermediate estimate[c] |
| LCC_KK_high | KK11[b] | high estimate[c] |

[a]Klein Goldewijk et al. (2010, 2011)
[b]Kaplan et al. (2011, 2012)
[c]This study; Sects. 2.5, 2.6, 2.7

The replacement of natural vegetation by crop and pasture leads to a decrease in SOA precursors, which is most likely responsible for the significant[3] changes in CDNC, LWP, and CRE for LCC_KK (Table S9; all cloud properties are grid means).

In our simulations, the surface albedo can either increase or decrease significantly depending on the season and the region. Deforestation can introduce brighter vegetation types and thus an increase in surface albedo (e.g. in Spain; Fig. 2a,b). However, especially after harvest, crop has a smaller canopy area fraction than natural vegetation. Thus, more of the soil, which is in some regions darker than the natural vegetation, is exposed to radiation. As a consequence, the annually averaged surface albedo decreases in parts of Europe (Fig. 2a,b) because the surface albedo there increases in summer but decreases in other seasons (not shown).

Land cover change also alters latent heat (LH) and sensible heat (SH) fluxes. The annual mean evaporative fraction (Evap_frac$=\frac{\text{LH}}{\text{SH}+\text{LH}}$; only calculated if both SH and LH are upward fluxes, otherwise set to zero) shows regionally no significant changes (Fig. 2c,d). In contrast, regional changes in the annual mean turbulent flux ($F_{\text{turb}} = \text{SH}+\text{LH}$; again only calculated if both SH and LH are upward fluxes) are significant for KK11 (Fig. 2f).

For HYDE11, the land surface temperature shows hardly any significant changes (Fig. 2g). For KK11, our results suggest that the changes in turbulent flux have a larger influence on the land surface temperature than the changes in surface albedo and evaporative fraction (Fig. 2). The decrease in turbulent flux regionally leads to a less efficient heat transport from the surface to the atmosphere, thus enhancing the land surface temperature (Fig. 2h). In contrast, the land use in Smith et al. (2016) induces a distinct cooling over Europe around AD 1. We think that the opposite signal is mainly caused by differences in the vegetation or atmospheric models; intercomparison studies have shown that not all models agree on the sign of annual mean temperature changes induced by deforestation over North America and in mid-latitudes (Lejeune et al., 2017; Winckler et al., 2018). In our simulations, significant warming occurs south of $40°$ N, which is in qualitative agreement with present-day satellite studies (Li

[3]The paired $t$-test was used. We chose a paired test since the natural fire emissions have the same interannual pattern across all simulations. The effect of multiple hypothesis testing was considered by controlling the false discovery rate as described in Wilks (2016) using $\alpha_{\text{FDR}} = 2 \cdot \alpha$.

et al., 2015, 2016d). These studies indicate that the impact of deforestation shifts from a warming in the tropics to a cooling in boreal regions, though with a high uncertainty in the exact latitude of transition (Li et al., 2015, 2016d).

## 3.2 Anthropogenic aerosol emissions around AD 100 and their magnitude compared to natural fire emissions

For the majority of scenarios and aerosol (precursor) species, the emissions from pasture burning contribute most to the total anthropogenic emissions (Fig. 3). The emissions from fuel consumption are smaller in most cases but on the same order of magnitude (Fig. 3). Averaged over the whole year, the emissions from crop residue burning are relatively small; however, in summer, the emissions are in some cases comparable to pasture burning and/or fuel consumption (e.g. Fig. 3b).

In the following, we compare the anthropogenic aerosol emissions to the natural fire emissions in AD 100; if the anthropogenic aerosol emissions are very low compared to the natural fire emissions, we would not expect any influence of the anthropogenic emissions. On the other hand, if the anthropogenic aerosol emissions are on the same order of magnitude as the natural fire emissions or higher, then they could have an impact on radiation and clouds, depending on the amount of natural aerosols from other sources (i.e. mineral dust, sea salt, organic aerosol emissions from biogenic sources, and sulfate from volcanic, oceanic, and terrestrial sources).

Next to the total anthropogenic aerosol emissions, Figs. 4, S2, S3 (for BC, OC, and $SO_2$, respectively) show also the natural fire emissions used in the simulations without anthropogenic aerosols (no_human, LCC_HYDE, LCC_KK; called "ref" in the figure) and the natural fire emissions for the respective scenarios where the natural fire emissions are set to 0 in areas where crop or pasture grow (LCC_HYDE_low, LCC_HYDE_int, LCC_KK_high; called "LCC" in the figure). By comparing "ref" to the sum of "LCC" and the anthropogenic emissions, the potential impact of the anthropogenic emissions can be estimated.

The anthropogenic emissions have a much less pronounced annual cycle than the natural emissions, the latter showing considerably higher emissions in summer than in winter. Averaged over the whole year, the anthropogenic emissions of the low (Fig. 4b) and the intermediate (Fig. 4c) scenarios are small compared to natural fire emissions ("ref"), while the anthropogenic emissions of the high scenario are similar. However, the comparison strongly depends on the season. For the low emission scenario, the total aerosol emissions from natural fires ("LCC") plus anthropogenic activities are obviously smaller than the natural fire emissions ("ref") in summer, i.e. humans overall reduce aerosol emissions. In contrast, the anthropogenic emissions are clearly higher than the natural fire emissions in winter, as illustrated in Fig. 4a, which shows the months January to March on a different scale (two orders of magnitude lower). For the intermediate emission scenario, the anthropogenic emissions are larger than the natural fire emissions ("ref") for approximately half of the year. The anthropogenic emissions for the high emission scenario are higher than the natural fire emissions for most of the year (Fig. 4d).

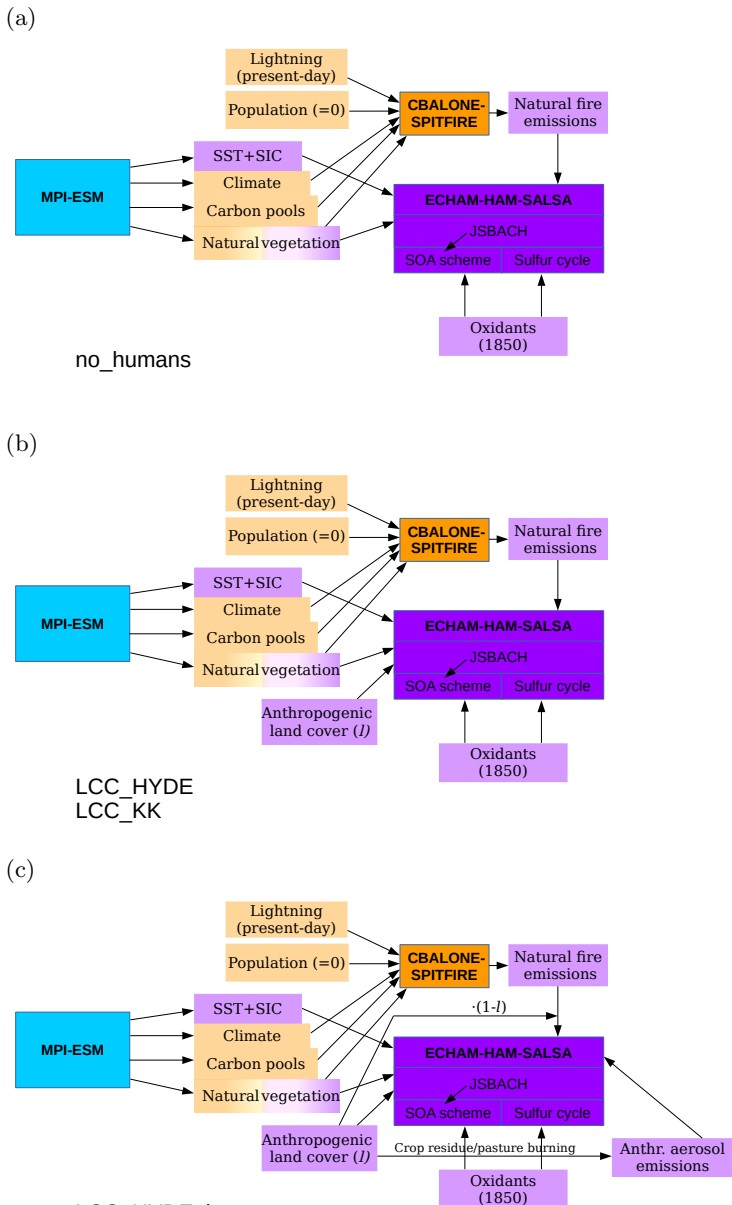

**Figure 1.** Illustrated are the setups of the 6 simulations conducted with ECHAM-HAM-SALSA (no_humans, LCC_HYDE, LCC_KK, LCC_HYDE_low, LCC_HYDE_int, and LCC_KK_high). Models that we used are shown in dark colours, whereas inputs to these models are shown in light colours. ECHAM-HAM-SALSA (violet) includes (among other components) the vegetation model JSBACH, a secondary organic aerosol scheme, and a sulfur cycle. Natural fire emissions were calculated with CBALONE-SPITFIRE (orange). For driving the two models, output from the Earth System Model MPI-ESM was used (blue; simulations come from the study by Bader et al., 2019, in review) among others.

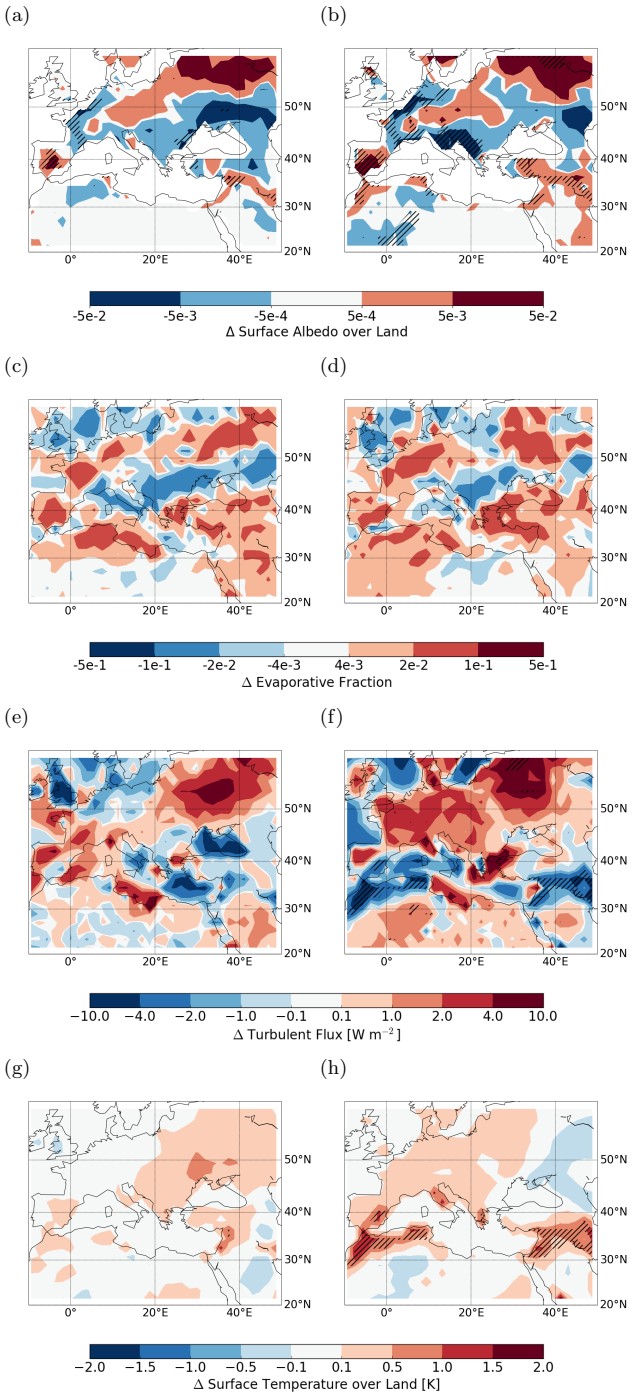

**Figure 2.** The impact of anthropogenic land cover change using HYDE11 (left) and KK11 (right). Shown are changes in surface albedo over land (a,b), evaporative fraction (c,d), turbulent flux (e,f), and land surface temperature (g,h). Statistically significant changes (5% significance level; $N = 20$) are hatched. Note that the SST in the simulations are fixed.

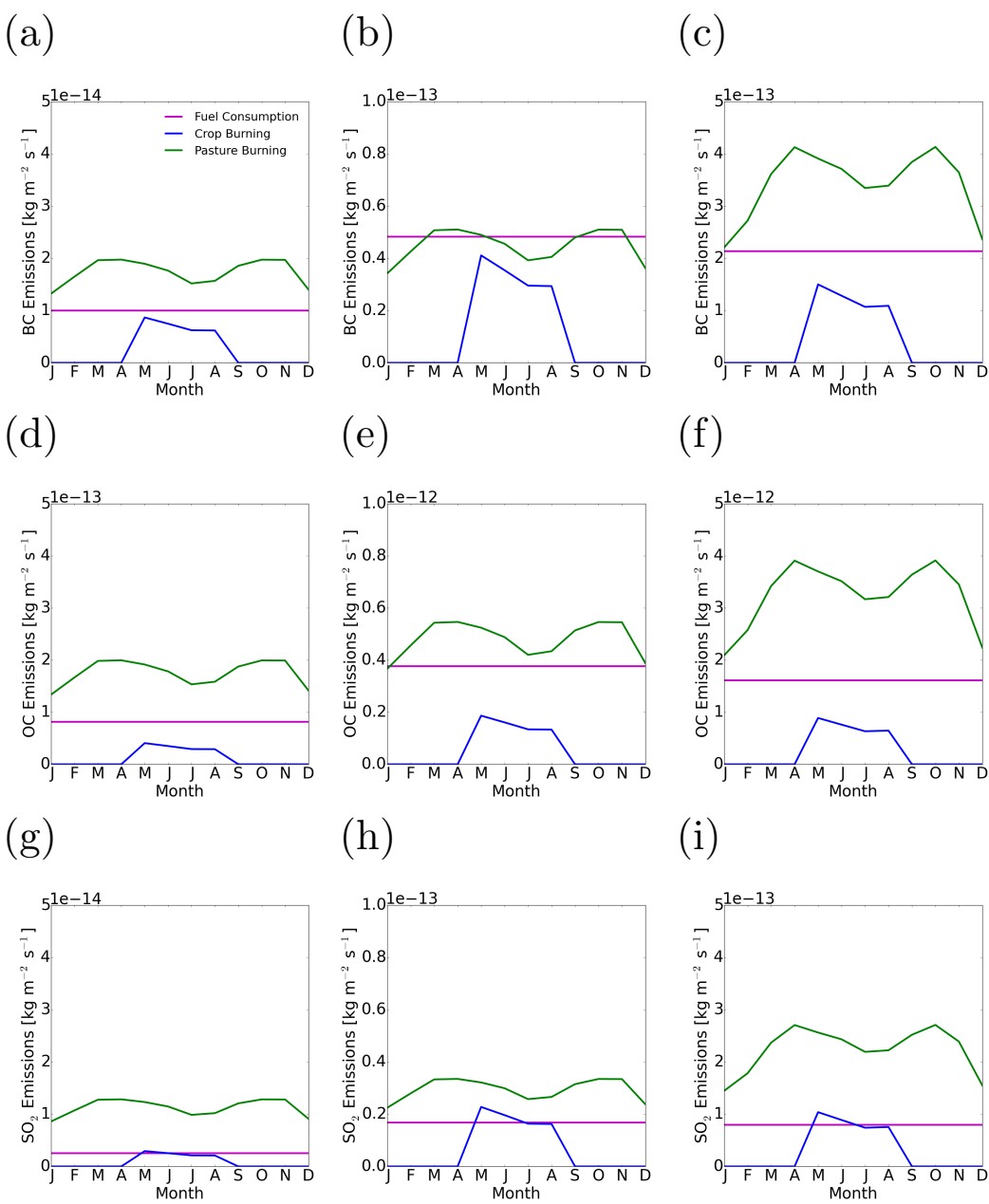

**Figure 3.** Monthly mean anthropogenic aerosol emissions due to fuel consumption, crop residue burning, and pasture burning in the study domain over the year. (a)-(c): BC; (d)-(f): OC; (g)-(i): $SO_2$ emissions. Left: low emission scenario; middle: intermediate emission scenario; right: high emission scenario. Note the different scales on the y-axis.

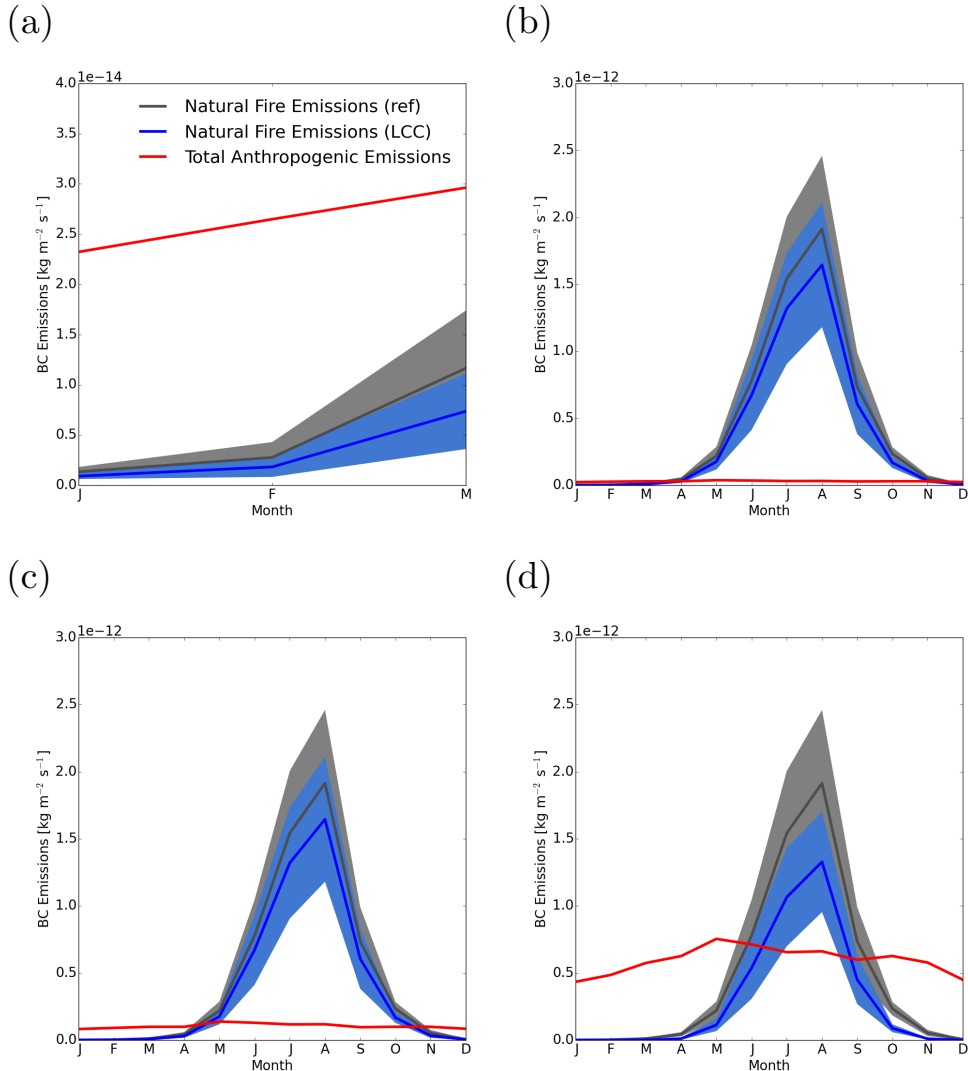

**Figure 4.** Monthly averaged BC emissions in our study domain for (a) the beginning of the year and (b)-(d) the whole year. The shadings show the standard deviation for each month over the 20-year period since our natural fire emissions have interannual variability. The grey line shows the natural fire emissions of the simulations no_human, LCC_HYDE, and LCC_KK, whereas the blue line shows the natural fire emissions of the simulations LCC_HYDE_low, LCC_HYDE_int, and LCC_KK_high, in which anthropogenic land cover change reduces the natural fire emissions. The red line shows the total anthropogenic aerosol emissions. In (a) and (b), the BC emissions from the low scenario are shown; note that the y-scale is smaller in (a) than in the other sub-figures. In (c) and (d), the emissions from the intermediate and the high scenarios are shown, respectively.

### 3.3 Comparison of anthropogenic aerosol emissions around AD 100 and AD 1850

To put our calculated anthropogenic aerosol emissions for AD 100 into some temporal context, we compare them to the anthropogenic aerosol emissions in AD 1850 based on the ACCMIP (Atmospheric Chemistry & Climate Model Intercomparison Project) inventory (Lamarque et al., 2010). The emissions are in both cases averaged over our study domain and the year. Anthropogenic ACCMIP emissions in AD 1850 include the sectors industry, land transport, maritime transport, residential and commercial combustion, agricultural waste burning on fields, and waste. Not all of these sectors produced aerosol emissions in AD 100; our emissions thus only include fuel consumption (both due to industrial as well as residential combustion), agricultural waste burning on fields, and pasture burning. Emissions from pasture burning are not an own ACCMIP sector; we expect that the anthropogenic ACCMIP emissions in AD 1850 could be somewhat higher if pasture burning were included.

The BC emissions from the high scenario in AD 100 are nearly as high as the emissions in AD 1850, whereas the emissions from the low and the intermediate scenarios are considerably lower (Fig. 5a). For OC, the emissions for the low and the intermediate scenarios are lower than the emissions in AD 1850, whereas the high scenario results in approximately twice as high aerosol emissions as in AD 1850 (Fig. 5b). The large OC emissions in AD 100 could be due to differences in emission factors: uncertainties in emission factors from biomass burning are large, and the composition of fuels was different in AD 1850 than in AD 100 (e.g. more coal in AD 1850). For $SO_2$, the emissions in AD 100 are for all scenarios clearly lower than those in AD 1850 (Fig. 5a), which might again be related to the larger contribution of fossil fuels in AD 1850.

Comparing the aerosol emissions per capita gives similar results (Fig. 6): relative to AD 1850, emissions are highest in AD 100 for OC, followed by BC and then $SO_2$. For OC, the per capita emissions are higher in AD 100 than in AD 1850 for the high and the intermediate scenarios. For BC, the per capita emissions are only higher for the high emission scenario, while the per capita emissions for $SO_2$ are lower than in AD 1850 for all scenarios.

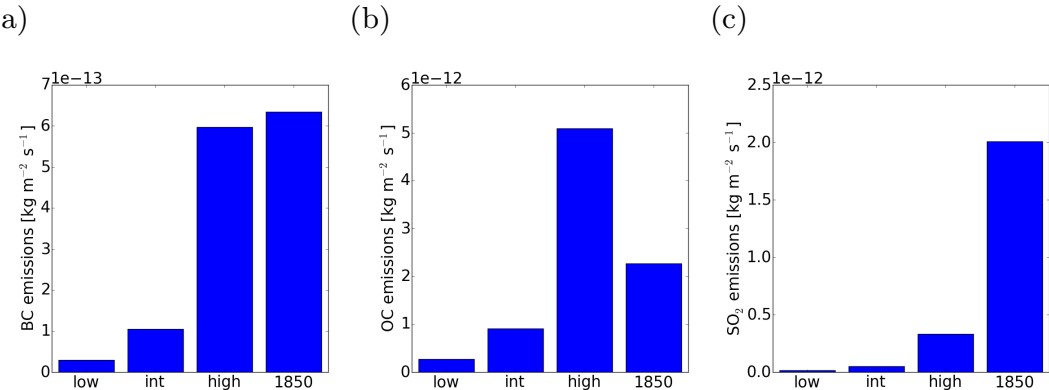

**Figure 5.** The annual mean anthropogenic aerosol emissions of (a) BC, (b) OC, and (c) $SO_2$ in AD 100 averaged over the study domain (10° W to 50° E, 20° N to 60° N) for the low emission scenario (low), the intermediate emission scenario (int), and the high emission scenario (high). Also shown are the anthropogenic aerosol emissions from the ACCMIP inventory for the year AD 1850 averaged over the same region.

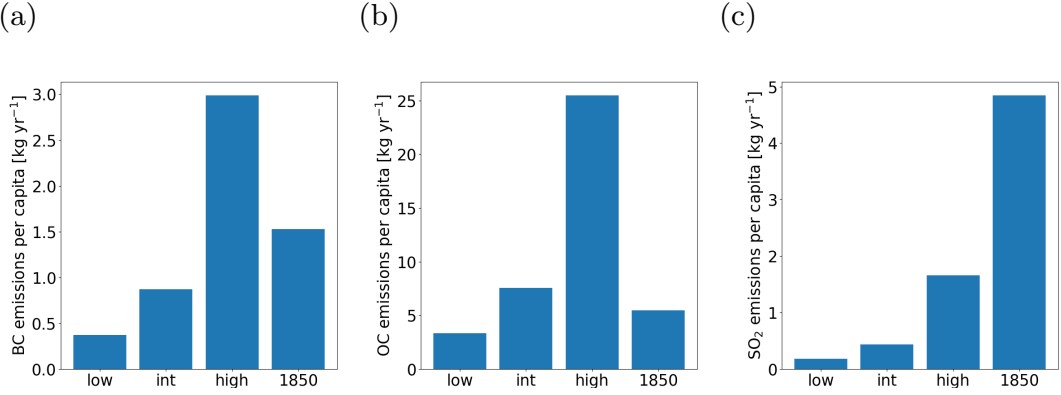

**Figure 6.** The same as Figure 5 but showing per capita emissions.

**Table 5.** Annual mean impact of anthropogenic aerosol emissions on different variables averaged over the study domain (10° W to 50° E and 20° N to 60° N): cloud droplet number concentration (CDNC), liquid water path (LWP), cloud cover (CC), aerosol radiative effect (ARE), and cloud radiative effect (CRE) for all simulations except no_human. Significant (5% significance level; $N = 20$) changes compared to the respective reference are marked with '*'. [1]: Changes are relative to HYDE11. [2]: Changes are relative to KK11.

| Var. | Unit | HYDE11 | KK11 | low[1] | | int[1] | | high[2] | |
|------|------|--------|------|--------|--------|--------|--------|--------|--------|
| CDNC | $10^9 \, \mathrm{m}^{-2}$ | 22.87 | 21.90 | 26.12 | $(+14.2\%)*$ | 32.76 | $(+43.2\%)*$ | 58.34 | $(+166.4\%)*$ |
| LWP | $\mathrm{g\,m}^{-2}$ | 47.64 | 45.98 | 56.69 | $(+19.0\%)*$ | 66.97 | $(+40.6\%)*$ | 84.47 | $(+84.1\%)*$ |
| CC | - | 0.483 | 0.480 | 0.496 | $(+2.7\%)*$ | 0.503 | $(+4.2\%)*$ | 0.522 | $(+8.7\%)*$ |
| ARE | $\mathrm{W\,m}^{-2}$ | $-0.66$ | $-0.64$ | $-0.60$ | $(-9.0\%)*$ | $-0.54$ | $(-18.0\%)*$ | $-0.52$ | $(-18.7\%)*$ |
| CRE | $\mathrm{W\,m}^{-2}$ | $-8.83$ | $-8.58$ | $-10.95$ | $(+23.9\%)*$ | $-13.01$ | $(+47.2\%)*$ | $-16.21$ | $(+88.9\%)*$ |

## 3.4 Impact of anthropogenic aerosol emissions around AD 100 on radiation and PM2.5 concentrations

In this section, the impact of anthropogenic aerosol emissions around AD 100 is assessed by comparing simulations considering anthropogenic land cover change and aerosol emissions with simulations that only consider anthropogenic land cover change (i.e. LCC_HYDE_low/LCC_HYDE_int compared with LCC_HYDE; LCC_KK_high compared with LCC_KK). The changes in some cloud properties and radiative effects, averaged over the study domain and the whole year, are shown in Table 5. Tables S10, S11, and S12 include more variables and show also seasonal averages for the low, the intermediate, and the high emission scenarios, respectively.

The BC and OC burdens (= vertically integrated mass) change significantly for all emission scenarios (Tables S10 to S12), with stronger changes for BC, whereas the $SO_4$ burden does not change significantly. This is due to the different emission sectors of the three aerosol types: while fire emissions are the only natural source of BC particles, OM also has biogenic sources, and $SO_4$ predominantly forms from gas-to-particle conversion of volcanic and oceanic precursor emissions. The changes in aerosol burdens show large seasonal differences: in summer, the BC and OC burdens decrease for the low and the intermediate emission scenarios because of the reduction in natural fires (Sects. 2.4, 3.2). In winter, the BC and OC burdens increase pronouncedly for all emission scenarios due to anthropogenic aerosol emissions.

Averaged over the study domain and the year, ARE changes by $0.06 \, \mathrm{W\,m}^{-2}$, $0.12 \, \mathrm{W\,m}^{-2}$, and $0.12 \, \mathrm{W\,m}^{-2}$ for the low, the intermediate, and the high emission scenarios, respectively (Table 5). Largest increases occur over North Africa and Europe (Fig. 7a-c). In contrast to ARE, the changes in CRE are negative and considerably more pronounced: $-2.11 \, \mathrm{W\,m}^{-2}$, $-4.17 \, \mathrm{W\,m}^{-2}$, and $-7.63 \, \mathrm{W\,m}^{-2}$ for the three scenarios, respectively. The changes predominantly occur over Europe (Fig. 7d-f). The cooling effect of clouds is enhanced because they become optically thicker and more abundant due to aerosol-induced increases in CDNC and LWP (Tables S10 to S12). Overall, the negative changes in CRE are larger than the positive changes in ARE. As a consequence, the anthropogenic aerosol emissions induce a decrease in land surface temperature (Fig. 7g-i).

Aerosol particles are also relevant because of their impact on human health. According to Makra (2015), air pollution had substantial consequences in cities in ancient times. As an example, the philosopher Seneca wrote that he needed to leave Rome

in order to escape the smoke and the kitchen smells and to feel better (Makra, 2015). As maintained by the World Health Organisation (WHO; 2006), the annual mean PM2.5 concentration should not exceed $10\,\mu g\,m^{-3}$. Figure 8 shows the background surface PM2.5 concentrations as well as the increases due to the anthropogenic emissions. For the low emission scenario, changes in the PM2.5 concentration are below $1\,\mu g\,m^{-3}$ everywhere. For the intermediate emission scenario, significant increases mainly range between 0.1 and $2\,\mu g\,m^{-3}$, which is for most places insufficient to increase the background concentrations to values above $10\,\mu g\,m^{-3}$. Only for the high emission scenario the increase in PM2.5 concentration is pronounced enough to exceed the WHO threshold in larger areas (e.g. Algeria, Egypt). The maximum PM2.5 concentrations are to some degree underestimated due to our coarse model resolution. As an example, Li et al. (2016c) found that the maximum simulated PM2.5 concentrations decrease by -21% if the resolution decreases from $0.5° \times 0.66°$ to $2° \times 2.5°$.

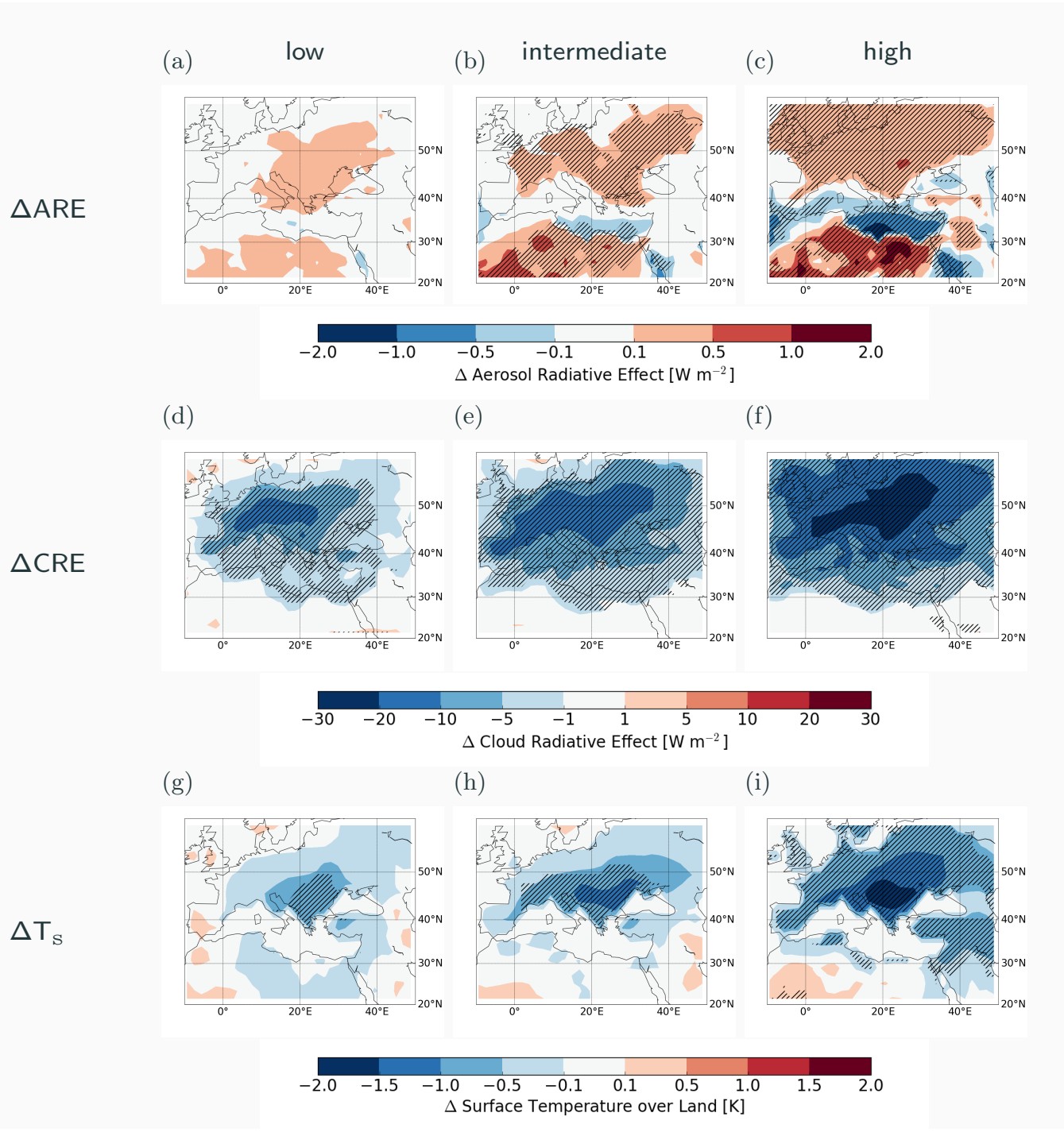

**Figure 7.** The impact of anthropogenic aerosol emissions for the low (left), the intermediate (middle), and the high emission scenarios (right). Shown are differences in aerosol radiative effect (top), cloud radiative effect (middle), and land surface temperature (bottom). Statistically significant changes (5% significance level; $N = 20$) are hatched. Note that the SST in the simulations are fixed.

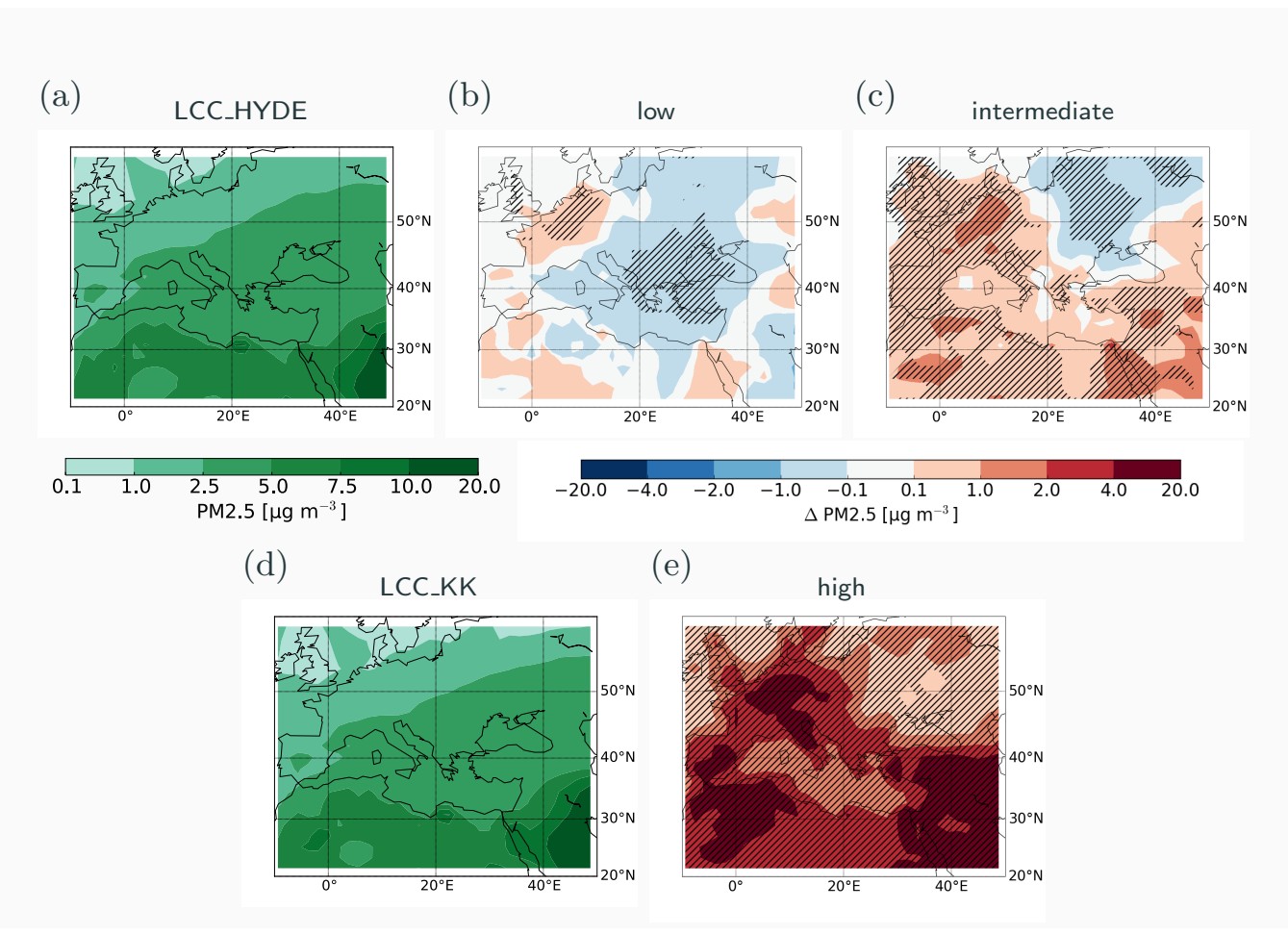

**Figure 8.** The background surface PM2.5 concentrations (a: LCC_HYDE; d: LCC_KK) and the changes in PM2.5 surface concentrations due to anthropogenic aerosol emissions for the low (b; LCC_HYDE_low), the intermediate (c; LCC_HYDE_int), and the high (e; LCC_KK_high) emission scenarios. Statistically significant changes (5% significance level; $N = 20$) are hatched.

## 4  Uncertainties and limitations

Because ECHAM-HAM-SALSA is an atmosphere-only model, SST and SIC were prescribed. Therefore, temperature does not react to a perturbation as much as with an interactive ocean, implying that, in the absence of attenuating feedbacks, our simulated changes in temperature should be underestimated. We repeated two of the simulations (no_human, LCC_HYDE_int) with a mixed-layer ocean (MLO) model having a depth of $50\,\mathrm{m}$ ($\approx 50$ years of simulation, half of it model spin-up), and arrived indeed at larger changes in land surface temperature (4 times stronger decrease in land surface temperature averaged over our study domain). Nevertheless, the temperature patterns occurring with fixed SST remain visible with the MLO setup: the land surface temperature decreases mainly over Europe due to anthropogenic aerosol emissions (Fig. S4). As expected, the aerosol-induced cooling is more widespread with the MLO, but it is basically limited to the Roman Empire. Since the changes in variables affecting temperature (e.g. CRE) are comparable for the different model setups, qualitatively similar, though stronger changes in the land surface temperature occur with the MLO. Nevertheless, we admit that a more detailed analysis using the MLO, including all scenarios, will provide more quantitative and perhaps partly different results. However, this is beyond the scope of this study.

The two land cover reconstructions that we chose differ widely and thus provide a measure for uncertainty. However, also the soil albedo and vegetation cover of the simulation no_human, which have a strong influence on the climate response, are subject to uncertainty. We found that the contribution of grasslands to the natural vegetation is quite high in our simulations. As an example, grasslands and tundra contribute $\approx 45\%$ to the natural vegetation between $0° \mathrm{E}$ and $30° \mathrm{E}$ and $30° \mathrm{N}$ and $60° \mathrm{N}$. If the forest fraction were underestimated in our simulation, the impact of the anthropogenic land cover change would likely also be underestimated. This would especially be the case for creating pasture, since grasslands are preferentially converted to pasture in the model (Reick et al., 2013). Furthermore, the crop phenology of JSBACH, which considers both winter and summer crop, allows two harvests of summer crops during one year in the extratropics depending on the heat sum. Generally, the productivity of crops was smaller in the Roman Empire than today, with fallow every two to three years. The number of harvests is thus likely overestimated in JSBACH (but not in the offline calculated crop residue burning emissions), which affects for example changes in surface albedo (which could be more positive in reality).

The anthropogenic aerosol emissions that we calculated are only rough estimates. To calculate aerosol emission factors for fuel consumption, we used measurements from present-day fireplaces and traditional stoves. The measurements show a large variability, which is caused e.g. by the fuel moisture, the burning device, the fire conditions (flaming versus smoldering), or the instrumental setup. To account for the large range of observed emission factors, we considered many measurements. Nevertheless, we cannot rule out that typical stoves used in the Roman Empire had systematically different emission factors than the majority of burning devices that we considered. Furthermore, the uncertainties of important parameters such as population size are considerable when going so far back in time. The emissions are thus highly uncertain, and it is extremely challenging to quantitatively verify them with palaeo records. At least there is indirect evidence for aerosol emissions associated with fuel consumption and agricultural burning: marble turned grey in antique towns due to smoke, and laws were introduced against air pollution (Makra, 2015). Moreover, pollen and charcoal records show high positive correlations between fire and crops,

weeds, and shrubs in Mediterranean and temperate regions in and around the Alps between 2300 BC and 800 AD (Tinner et al., 2005, 2009). Palaeo records further suggest that controlled burning was used to introduce and establish sweet chestnut in some regions (Morales-Molino et al., 2015).

Another simplification is our lack of spatial and temporal variations: for many variables, we estimated one "typical" value for the whole Roman Empire, and our anthropogenic emissions show no trends over the years (e.g. caused by wars). The time around AD 100 was a relatively stable period, characterised by expansion of infrastructure, economic wealth, and quite low military activities. Nevertheless, the emissions were more dynamic in reality than in our simulations, e.g. due to Trajan's Dacian Wars (AD 101/102 and AD 105/106), which caused population movements as well as a possible change in anthropogenic aerosol emissions (e.g. regional emissions associated with warlike activities such as burning of villages).

The impact of our simulated anthropogenic aerosol emissions strongly depends on the natural background and its seasonality. In our simulations, oceanic DMS concentrations, dust potential sources, and volcanic tropospheric $SO_2$ emissions are representative for present-day conditions, which could have an impact on the background aerosol concentrations. Moreover, fire models show large differences, and thus it is unclear how realistic the fire emissions and the strong seasonal cycle of our CBALONE-SPITFIRE simulations (Sect. 3.2) are. The fire emissions from our simulations are one order of magnitude higher than the emissions by van Marle et al. (2017) in the study domain (Sect. S9), indicating that our natural fire emissions could be overestimated. On the one hand, this could indiciate that our background aerosol concentration is potentially overestimated as well. On the other hand, our model neglects pure biogenic aerosol nucleation and nitrate aerosols. Using the GLOMAP model, Gordon et al. (2016) have shown that the global annual mean CCN concentration (at 0.2% supersaturation and at cloud base level) was 12% higher in the pre-industrial atmosphere when pure biogenic nucleation is considered. The effect is not the same over the year since terpene emissions are larger in summer (Gordon et al., 2016). Therefore, our simulated background concentration when we see the highest impact of anthropogenic aerosols (winter, spring) might not be affected too much by the neglect of biogenic nucleation.

The aerosol background does not only depend on emissions, but also on other parameters such as the simulated size distribution or the calculation of removal processes. A simplification of our study is that most of the tropospheric chemistry is not calculated, hence we prescribe the oxidants that are important for the sulfur chemistry and the SOA formation. Also the vertical distribution of the aerosol particles is essential when their effect on radiation and clouds is analysed. In ECHAM-HAM, vertical mixing is strong in the lower troposphere (Veira et al., 2015), and thus the anthropogenic aerosol particles emitted near the surface can easily reach altitudes where clouds form. If this vertical mixing were overestimated in the model, then the aerosol-cloud interactions would likely be overestimated as well.

Last but not least, aerosol-radiation and aerosol-cloud interactions from aerosol-climate models are uncertain. As an example, some studies show that the models typically overestimate the effect of aerosols on the cloud liquid water content, at least in some regions (Bender et al., 2018). The aerosol effective radiative forcing also depends on the minimum CDNC value (Hoose et al., 2009); since we lowered the minimum CDNC to $1\,\mathrm{cm}^{-3}$, the aerosols have a stronger impact on radiation than with the standard setup of ECHAM-HAM-SALSA where the minimum CDNC is $40\,\mathrm{cm}^{-3}$. We conducted additional simulations showing that the total ERF due to aerosols between AD 1850 and AD 2000 is $\approx -2.4\,\mathrm{W\,m}^{-2}$ with our decreased minimum

CDNC value. Our $\text{ERF}_{\text{ari+aci}}$ thus lies at the lower end of model estimates and is outside the range of IPCC's expert judgment ($-1.9$ to $-0.1\,\text{W}\,\text{m}^{-2}$; Boucher et al., 2013). Nevertheless, we expect that the anthropogenic aerosol emissions around AD 100 would still increase the CDNC in the Roman Empire (and thus induce a cooling effect) with the standard minimum CDNC value of $40\,\text{cm}^{-3}$: the simulated CDNCs are above $40\,\text{cm}^{-3}$ in the lower troposphere for all seasons when averaged over our

study domain (not shown). The strong total $\text{ERF}_{\text{ari+aci}}$ with a minimum CDNC of $1\,\text{cm}^{-3}$ therefore mainly results from other regions, e.g. the Arctic and the Antarctic.

## 5   Conclusions

The hypothesis of this study was that anthropogenic activities associated with land cover and aerosols already had a noticeable influence on some climate variables in the Roman Empire around AD 100. To test this hypothesis, we first created three different

scenarios of anthropogenic aerosol emissions for the Roman Empire. These scenarios were combined with two existing land use datasets.

     Simulations with an aerosol-enabled global climate model showed that the anthropogenic land cover reconstruction by KK11 (Kaplan et al., 2011, 2012) induces significant decreases in turbulent flux and thus a warming in parts of North Africa and the Middle East, whereas the reconstruction based on HYDE11 (Klein Goldewijk et al., 2010, 2011) has nearly no impact. Mainly

over Central and Eastern Europe, anthropogenic aerosol emissions lead to an enhanced cooling effect of clouds (i.e. a more negative cloud radiative effect), the extent and strength of which depends on the different scenarios. Our model likely overestimates changes in the cloud radiative effect to some degree due to the lowered minimum CDNC (Hoose et al., 2009). Overall, land use thus generally increases the land surface temperature in our simulations, whereas anthropogenic aerosol emissions decrease it. Since we prescribe the SST, our simulated changes in temperature are strongly underestimated. Simulations with

an interactive ocean are therefore needed for a more quantitative analysis.

     Around AD 100 temperatures in Europe (as well as China and the Northern Hemisphere in general) were warmer than the average over the last two millenia (PAGES 2k Consortium, 2013; Ge et al., 2013; Christiansen and Ljungqvist, 2012), an era that has been called the "Roman Warm Period". While our results imply that anthropogenic land cover change may have regionally contributed to this warming, aerosol-cloud interactions would have attenuated it, suggesting other causes of the

Roman Warm Period, e.g. ocean dynamics or solar forcing.

     Our scenarios show that pasture burning could have been an important source of aerosol particles. A better understanding of the processes that drive the frequency, the seasonality, and the emissions of pasture burning could therefore be essential to quantify the anthropogenic impact far back in time. These processes could be implemented in a vegetation-fire model that is coupled with an aerosol model. Simulations with a higher spatial resolution would moreover allow to account for the large

regional variations in aerosol emissions within the Roman Empire – information which can be provided by archaeologist and historians. Therefore, further work should include collaborations with them in order to incorporate better understanding of human behaviour in Classical Antiquity, particularly on fuel consumption and the timing and extent of the use of fire. Such

collaborations could also allow to assess the effect of economic crises (e.g. in the third century AD) and wars on aerosol emissions.

*Code availability.* TEXT

*Data availability.* The data can be found at: https://data.iac.ethz.ch/Gilgen_et_al_2019_RomanEmpire/

5 *Code and data availability.* TEXT

*Sample availability.* TEXT

*Video supplement.* TEXT

**Appendix A**

**A1**

10 *Author contributions.* AG had the initial idea for the study, calculated the anthropogenic aerosol emissions, adapted the anthropogenic land cover data, conducted the simulations with ECHAM-HAM-SALSA, analysed the results, created the figures, and wrote the main part of the manuscript. SW did the major work for calculating the CBALONE-SPITFIRE emissions. JOK developed the KK11 land use reconstruction and contributed to the development of the anthropogenic emission scenarios. TK implemented the VBS in ECHAM-HAM-SALSA that was used in this study. UL was responsible for supervising AG. All authors were involved in discussions about the setup of the simulations and 15 the study; all authors participated in the analysis and the writing.

*Competing interests.* The authors declare that no competing interests are present.

*Disclaimer.* TEXT

*Acknowledgements.* This work was supported by a grant from the Swiss National Science Foundation (SNF) for the Sinergia project "Paleo fires from high-alpine ice cores" (CRSII2_154450) and by a grant from the Swiss National Supercomputing Centre (CSCS) under project ID s652. We are deeply grateful to Thomas Raddatz, who provided temporally high resolution data of MPI-ESM to drive CBALONE-SPITFIRE. He also helped the main author with questions about JSBACH and its anthropogenic land cover scheme. We thank Robyn Veal, Edouard Davin, Luisa Ickes, Silvia Kloster, Tanja Stanelle, and David Neubauer for valuable inputs and discussions. Furthermore, we thank Jürgen Bader for providing SST and SIC of at that time unpublished MPI-ESM simulations and Silvia Kloster and Christian Reick for enabling collaborations between people from the MPI Hamburg and ETH Zürich. We generally thank the developers of the ECHAM-HAMMOZ model, which is developed by a consortium composed of ETH Zürich, Max Planck Institut für Meteorologie, Forschungszentrum Jülich, University of Oxford, the Finnish Meteorological Institute, and the Leibniz Institute for Tropospheric Research and is managed by the Center for Climate Systems Modeling (C2SM) at ETH Zürich. We also acknowledge all the people that made their published data available, e.g. the developers of CESM2.0 WACCM and the people contributing to CMIP6 input data and to the HYDE database. Last but not least, we cordially thank the two anonymous reviewers for their well-conceived comments and helpful suggestions.

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
