# Peer review of "Effects of land use and anthropogenic aerosol emissions in the Roman Empire"

_Climate of the Past, 2019_

## Referee Comment (RC1) · Anonymous Referee #1 · 15 Jul 2019

It was a pleasure to read the paper entitled "Did the Roman Empire affect European climate? A new look at the effects of land use and anthropogenic aerosol emissions".

The paper focuses on the impact of the Ancient Romans around 100 AD. This period corresponds to the maximum expansion of the Roman empire, under the Emperor Traiano, and was considered by the authors as representative of a potential past anthropogenic impact on the climate system. The authors evaluate the anthropogenic effect of land use and aerosol emissions.

In my view, the paper is really well written, properly structured and easy to read.

The authors properly discussed the assumptions introduced that provided the uncer-

tainties in the model.

Perhaps, a more detailed archaeological discussion would further increase the quality of this paper. I was wondering, for example, what is the potential influence of Roman battles in your model, where thousands of people moved in Europe (e.g. during the Dacian War), burned villages, re-worked metals and so on. This impact should be intense, and maybe influenced significantly the population size and distribution in short times. In my view, the paper meets the standards of this journal and I have nothing against its publication in cp.

Please correct Pompeji (it sounds German to me) (p.7 L. 28).

―――――――――――――――――――

---

## Referee Comment (RC2) · Anonymous Referee #2 · 31 Jul 2019

The manuscript by Gilgen and co-authors describes a series of climate model simulations during the peak of the Roman Empire, in which they study the role of anthropogenic activities on climate at the continental scale. The manuscript is very well written, with ample information that explain the experiments, the data sources, the assumptions made, and the associated uncertainties. Had this been a more technical paper describing the method only, I would have had only trivial comments to make. However, since the goal of the paper is beyond the construction of a model, and tries to explain (with uncertainties) the anthropogenic influence on climate during the Roman era, I am afraid that I have more substantial comments.

[Figure]

Major comments

The discussion on climate is fairly thin, compared to the bulk work presented in the manuscript. Granted, a number of metrics are being provided, but no in-depth analysis is provided, at least nowhere near the technical description of the model. For example, terms like evaporative fluxes, turbulent fluxes, precipitation, cloud cover, liquid water path, are simply presented but never analyzed. In a climate-focused paper, I would expect a much deeper analysis of these results, and discussions on how they influence each other, and how each one is affecting climate. Other examples of interest might be how precipitation changes between simulations might lead to drought frequency changes, how cloud cover can alter lightning (important for fires), etc. Last but not least, I was not convinced that the anthropogenic influence 2000 years ago did indeed impact climate significantly, as was promised in the abstract. The paragraph in page 26, around line 30, was a strong contributor to this.

There is no discussion on how uncertainties of the rest of the world affect results. I would expect that these uncertainties would be at least as large as in Europe, if not larger. How do these affect the base model climate over the region, via teleconnections and long-range transport? The authors correctly claim (and cite relevant literature) that the quantification of the anthropogenic impact on present-day climate depends on the choice of the preindustrial year of reference, and on whether this is 1750 or 1850. Why they don't explore this in their model for this study via rest-of-the-world influences? No sophisticated analysis would be required, just a ballpark halving/doubling of the global emissions.

As clearly stated a couple of times in the manuscript, surface air temperature is not a good metric, due to the model design (fixed SSTs). Still, the authors frequently refer to it, and even present several results and a figure about it. I would strongly recommend to not do so, since what they get is a muted response due to the fixed SST assumption, and the fact that they present surface temperatures over land only does not change this fact. This is evident when they used a mixed-layer ocean, and the signal increased 5-

fold. As a matter of fact, since the model is capable of simulating a mixed-layer ocean, and the authors did a couple of experiments with it, why not do all of them with such a configuration?

The supplementary material contains a lot of very useful and interesting information. I found that almost the whole supplementary material can fit into a standalone technical paper, and then the main manuscript can be a climate-focused paper. Then tables S8-S12, which contain a lot of numbers which remained virtually uncommented in the text, can be promoted to the main body instead of the supplement, where they belong, at least in my opinion.

Specific comments (please read e.g. 3.30 as page 3, line 30)

Throughout the manuscript, the years reported (e.g. AD 1, AD 10, AD 100) are described as literally *those* years. Instead, I believe the authors mean that these are climatological means around those years, and not that e.g. AD 1 is different from AD 2. This should be rephrased across the manuscript.

The authors do acknowledge their attempt to construct consistent scenarios when modifying model parameters for the low/intermediate/high estimates. However, it is not clear whether they checked that crop yield (and the implied animal husbandry from pasture lands) consistently provides the necessary food to feed the different populations across the scenarios, or the food chain is broken so that either too little or too much food is being produced. A comment on this would be appreciated.

3.30: "were reduced" needs to be expanded in the main text, right now this information only exists in S3.

Page 6, paragraph around line 5: The differences between the model-used data and the original estimates is very large. I was very much surprised by the values up to 41%. Is this really due to different datasets and regridding? Which assumptions introduce the largest changes from original to model-used estimates?

8.5-8: Given the large differences in emissions factors for heating (presented in 7.32-33) isn't this a gross oversimplification?

8.20: I guess another important assumption made here is that the ratio between free/enslaved people also remained the same?

Page 11, "over 20 years" (line numbering is off there): Is this statement referring to a climatological analysis, or a transient simulation?

15.9-12: How much lower were the CDNCs in the AD 100 simulation compared to preindustrial? Also, what are other studies do in terms of the lower threshold of CDNC, for studies where humans were irrelevant (e.g. LGM)?

15.16 and 19.8: I believe S8-12 contain useful information and they should not be in the supplement. Their discussion should be largely expanded as well.

24.6: "5 times stronger" refers to values over land only?

Page 25, first half: Why discuss so much about nitrate aerosols, which were not included in the study?

26.21: Please add "strongly" in front of "underestimated".

Section S3 mentions interannual variability. Is this in a transient or climatological sense?

Section S4, page 4, middle: Why not scale a_n with population density or crop use (or both)? Even an empirical scaling would have been better than a constant value with time, something like what was done with the vegetated area per gridbox.

Section S4, page 4, middle: "over 1830 and 1840" means literally these years, or a climatology from 1830 to 1840, or something else?

Technical corrections

Page 1, line 6: Please explain HYDE and KK11 in the abstract.

6.13: Is "e.g." correct, or "i.e." was meant to be there? In either case, this sounds like a vague statement, where a solid reference should be provided for the model description.

12.32: "Sect. S7" should had been "Sect. S6"? S7 seems irrelevant there.

16.7: Please add a comma after "harvest".

24.5: What is the depth of the mixed-layer ocean?

24.20: Please fix typo "smoldering".

Table S1: please change "degree" to "degrees".

Table S2 legend: Please define what is meant by "some" aerosol emissions.

Section S3 is very interesting and important, similar to section 2.5 in the main text, I feel it belongs to the main manuscript text.

Section S3, line 4: please define what is meant by "some" selected 30-year periods.

---

## Author Comment (AC1) · 30 Aug 2019

*We are glad that the reviewer enjoyed reading our paper and thank for his/her suggestions. Our answers are italicised.*

It was a pleasure to read the paper entitled "Did the Roman Empire affect European climate? A new look at the effects of land use and anthropogenic aerosol emissions". The paper focuses on the impact of the Ancient Romans around 100 AD. This period corresponds to the maximum expansion of the Roman empire, under the Emperor Traiano, and was considered by the authors as representative of a potential past

anthropogenic impact on the climate system. The authors evaluate the anthropogenic effect of land use and aerosol emissions. In my view, the paper is really well written, properly structured and easy to read. The authors properly discussed the assumptions introduced that provided the uncertainties in the model. Perhaps, a more detailed archaeological discussion would further increase the quality of this paper. I was wondering, for example, what is the potential influence of Roman battles in your model, where thousands of people moved in Europe (e.g. during the Dacian War), burned villages, re-worked metals and so on. This impact should be intense, and maybe influenced significantly the population size and distribution in short times.

*Thank you for this interesting suggestion. We agree that the quality of the paper improves by including these aspects. The following changes were made:*

- *In Section "Uncertainties and Limitations", we briefly discuss the influence of wars: **"Another simplification is our lack of spatial and temporal variations: for many variables, we estimated one "typical" value for the whole Roman Empire, and our anthropogenic emissions show no trends over the years (e.g. caused by wars). The time around AD 100 was a relatively stable period, characterised by expansion of infrastructure, economic wealth, and quite low military activities. Nevertheless, the emissions were more dynamic in reality than in our simulations, e.g. due to Trajan's Dacian Wars (AD 101/102 and AD 105/106), which caused population movements as well as a possible change in anthropogenic aerosol emissions (e.g. regional emissions associated with warlike activities such as burning of villages)."***

- *The last paragraph of "Conclusions" was extended: **"Simulations with a higher spatial resolution would moreover allow to account for the large regional variations in aerosol emissions within the Roman Empire – information which can be provided by archaeologist and historians. Therefore, further***

*work should include collaborations with them in order to incorporate better understanding of human behaviour in Classical Antiquity, particularly on fuel consumption and the timing and extent of the use of fire. Such collaborations could also allow to assess the effect of economic crises (e.g. in the third century AD) and wars on aerosol emissions."*

- *In Section "Fuel consumption per capita", we shortly mention the impact of glass recycling:* ***"Note that calculating emissions for individual sectors can be very challenging; as an example, recycling of glass was common (Stern 1999; Freestone 2015), which needs to be considered when estimating fuel consumption associated with glass making."***

In my view, the paper meets the standards of this journal and I have nothing against its publication in cp. Please correct Pompeji (it sounds German to me) (p.7 L. 28).

*We corrected it.*

---

## Author Comment (AC2) · 30 Aug 2019

*We cordially thank the reviewer for his/her careful reading, the well-conceived comments, and the helpful suggestions. Our answers are italicised.*

**Major comments**

The discussion on climate is fairly thin, compared to the bulk work presented in the manuscript. Granted, a number of metrics are being provided, but no in-depth analysis is provided, at least nowhere near the technical description of the model.

[Figure]

For example, terms like evaporative fluxes, turbulent fluxes, precipitation, cloud cover, liquid water path, are simply presented but never analyzed. In a climate-focused paper, I would expect a much deeper analysis of these results, and discussions on how they influence each other, and how each one is affecting climate. Other examples of interest might be how precipitation changes between simulations might lead to drought frequency changes, how cloud cover can alter lightning (important for fires), etc. Last but not least, I was not convinced that the anthropogenic influence 2000 years ago did indeed impact climate significantly, as was promised in the abstract. The paragraph in page 26, around line 30, was a strong contributor to this. There is no discussion on how uncertainties of the rest of the world affect results. I would expect that these uncertainties would be at least as large as in Europe, if not larger. How do these affect the base model climate over the region, via teleconnections and long-range transport? The authors correctly claim (and cite relevant literature) that the quantification of the anthropogenic impact on present-day climate depends on the choice of the preindustrial year of reference, and on whether this is 1750 or 1850. Why they don't explore this in their model for this study via rest-of-the-world influences? No sophisticated analysis would be required, just a ballpark halving/doubling of the global emissions. As clearly stated a couple of times in the manuscript, surface air temperature is not a good metric, due to the model design (fixed SST). Still, the authors frequently refer to it, and even present several results and a figure about it. I would strongly recommend to not do so, since what they get is a muted response due to the fixed SST assumption, and the fact that they present surface temperatures over land only does not change this fact. This is evident when they used a mixed-layer ocean, and the signal increased 5-fold. As a matter of fact, since the model is capable of simulating a mixed-layer ocean, and the authors did a couple of experiments with it, why not do all of them with such a configuration? The supplementary material contains a lot of very useful and interesting information. I found that almost the whole supplementary material can fit into a standalone technical paper, and then the main manuscript can be a climate-focused paper. Then tables S8-S12, which contain a lot

of numbers which remained virtually uncommented in the text, can be promoted to the main body instead of the supplement, where they belong, at least in my opinion.

*We understand the reviewer's point of view and agree that the climate analysis is expected to be deeper considering the title and the abstract of the paper. Our main interest was whether changes in anthropogenic land cover and aerosol emissions had an impact on some climate variables around AD 100. We admit that we were especially interested in how the land surface temperature changed and thus focused in our analysis on variables that could lead to such changes (e.g. changes in surface albedo, turbulent flux, or aerosol/cloud radiative effects). However, we fully agree with the reviewer that the focus on surface temperature is somehow problematic because of the model setup with fixed SST. We conducted simulations with fixed SST because the simulations take a lot of (computational) time (mainly because of the sectional aerosol model including secondary organic aerosols) and simulations with fixed SST are considerably shorter than those with a mixed-layer ocean model since a much longer spin-up is needed to equilibrate even a mixed-layer ocean model. For the same reason, we were limited in the number of simulations that we conducted. Furthermore, one anyhow first needs simulations with fixed SST to drive the mixed-layer ocean model. Out of curiosity, we then repeated two of the simulations with the mixed-layer ocean model to see how different the results are. As mentioned in the paper, the surface temperature showed stronger and more widespread changes with the mixed-layer ocean model. Nevertheless, the changes in surface temperature are qualitatively similar (see new Supplementary Fig. S4): a regionally very limited warming in Syria likely due to the changes in land cover and, more importantly, a more distinct and more widespread cooling in Europe due to the changes in aerosols. (Since we only calculated anthropogenic aerosol emissions for the Roman Empire, the cooling effect of the aerosols is mainly restricted to the study domain.) We thus believe that our simulations with fixed SST have some value and can provide information about the approximate location and sign of the expected surface temperature change, although we agree that the result is*

*rather qualitative than quantitative. We wrote a paragraph in the Section "Uncertainties and Limitations" about this.*

*Considering that the climate analysis is restricted to some variables and in the case of land surface temperature of qualitative nature, we followed the reviewer's suggestion and shifted the focus of this paper somewhat more to the technical aspects, i.e. the method and the experimental setup. As a consequence, we rephrased the title, the abstract, and part of the conclusions. Moreover, we integrated a large part of the Supplementary Material to the main manuscript. The results of the simulations with the fixed SST are now shortened and it is stated that more simulations using a mixed-layer ocean model are needed for more quantitative results. We now refrain from showing specific numbers of changes in surface temperature in the text and the tables. As a consequence, the section "Combined climate impact of anthropogenic land cover change and aerosol emissions" was deleted. However, we still leave some figures showing changes in surface temperature for qualitative purposes (e.g. to show that the changes in turbulent heat fluxes induce a local warming for KK11). In a follow-up paper including more simulations with the mixed-layer ocean model, we have then the possibility to have a deeper look at climate processes, including e.g. changes in precipitation and teleconnections as suggested by the reviewer.*

*We agree that the paragraph on page 26 sounds like the anthropogenic aerosols only have an effect due to a possible underestimation in natural aerosols and the lowered minimum CDNC. However, we expect that the anthropogenic aerosol emissions would still have a cooling effect (though a reduced one) with the standard CDNC minimum threshold of ECHAM-HAM ($40\ cm^{-3}$) because the CDNC background concentrations in our study region are between 100 and $200\ cm^{-3}$ in the lower troposphere (averaged over time and the study domain for the simulation no_human) and the absolute increases in CDNC are rather pronounced, especially for the high emission scenario (roughly $100\ cm^{-3}$). We therefore rephrased the abstract and added the following paragraph to Section "Uncertainties and limitations": **"Nevertheless, we expect that***

*the anthropogenic aerosol emissions around AD 100 would still increase the CDNC in the Roman Empire and thus induce a cooling effect with the standard minimum CDNC value of 40 cm$^{-3}$: the simulated CDNCs are above 40 cm$^{-3}$ in the lower troposphere for all seasons when averaged over our study domain (not shown). The strong total ERF$_{ari+aci}$ with a minimum CDNC of 1 cm$^{-3}$ therefore mainly results from other regions, e.g. the Arctic and the Antarctic."*

**Specific comments** (please read e.g. 3.30 as page 3, line 30)

- Throughout the manuscript, the years reported (e.g. AD 1, AD 10, AD 100) are described as literally *those* years. Instead, I believe the authors mean that these are climatological means around those years, and not that e.g. AD 1 is different from AD 2. This should be rephrased across the manuscript.

- *The reviewer is correct. Some data that we used has a low temporal resolution since the uncertainties are large. As an example, the data from the HYDE database has a resolution of 100 years. This data is therefore indeed representative for the period around, for example, AD 100. However, when we selected one "year" from such a database, we refer to this specific year (e.g. "AD 100") to make our method comprehensible and also to match the data as presented in the database exactly. Other data, e.g. the output of the simulations, is indeed always referring to climatological means. We went through the paper again and rephrased the sentences when necessary.*

- The authors do acknowledge their attempt to construct consistent scenarios when modifying model parameters for the low/intermediate/high estimates. However, it is not clear whether they checked that crop yield (and the implied animal husbandry from pasture lands) consistently provides the necessary food to feed the different populations across the scenarios, or the food chain is broken so that

either too little or too much food is being produced. A comment on this would be appreciated.

- *Concerning the crop yield, we considered the population size to assure that the food chain is not broken. We now explicitly mention this in the text. When estimating pasture burning, the link between population and the pasture burning emissions was less strong. We used the HYDE reconstruction with the lower pasture area for the low and the intermediate population size, while we used the KK11 reconstruction with the higher total pasture area for the scenario with the highest population size. This results in pasture areas per capita between 0.58 ha and 1.0 ha for the different emission scenarios, which is in reasonable agreement with other studies; the values are very similar to Goldewijk et al. (2011, 2017) and somewhat lower than in Weiberg et al. (2019) (1.75 ha). We now mention these numbers in the paper.*

- 3.30: "were reduced" needs to be expanded in the main text, right now this information only exists in S3.

- *As suggested by the reviewer, we moved Sect. S3 from the supplementary material to the main manuscript.*

- Page 6, paragraph around line 5: The differences between the model-used data and the original estimates is very large. I was very much surprised by the values up to 41%. Is this really due to different datasets and regridding? Which assumptions introduce the largest changes from original to model-used estimates?

- *We apologise for having made a mistake in the text, for KK11 the underestimation is even 47% instead of 41%. Part of the underestimation is related to the binary land sea mask. Part of the crop and pasture areas lie in locations that are ocean grid points in our model, where of course no vegetation can grow. The much larger underestimation for KK11 compared to HYDE11 is however related*

[Figure]

*to areas that are subject to anthropogenic land use in the KK11-reconstruction, but not hospitable to plants in the model because they are desert (e.g. large part of the Arabian Peninsula). We added these explanations to the paper.*

- 8.5-8: Given the large differences in emissions factors for heating (presented in 7.32- 33) isn't this a gross oversimplification?

- *This is true. We changed the text to:* **"Although more fuel for heating was generally consumed where and when it was cold (Malanima et al. 2006, Warde et al. 2006), we assume a constant fuel consumption over the year and over latitudes since we do not differentiate between heating, cooking, iron production, and other burning activities in our calculation. ... Given the large uncertainties about the relative importance of residential heating to total fuel consumption, this seems justified as a first order approximation."**

- 8.20: I guess another important assumption made here is that the ratio between free/enslaved people also remained the same?

- *The ratio between free people and slaves needs indeed also to be considered. We now mention this in the paper.*

- Page 11, "over 20 years" (line numbering is off there): Is this statement referring to a climatological analysis, or a transient simulation?

- *Climatological; we added this for clarification.*

- 15.9-12: How much lower were the CDNCs in the AD 100 simulation compared to preindustrial?

- *In our simulations, the CDNC burdens in the Roman Empire in AD 100 are between 2.6 and $5.8 \cdot 10^{10}$ $m^{-2}$ (depending on emission scenario). In our preindustrial simulation, it is $6.0 \cdot 10^{10}$ $m^{-2}$. On the global average, the values were*

*between 3.1 and $3.2 \cdot 10^{10}$ m$^{-2}$ for AD 100 and $3.6 \cdot 10^{10}$ m$^{-2}$ for preindustrial (we didn't estimate anthropogenic emissions outside the Roman Empire for AD 100). This highlights that using a high CDNC minimum threshold would generally have a larger effect in our AD 100 simulations than in our preindustrial simulations. The threshold has the largest effect in the polar regions (Arctic and Antarctic).*

- Also, what are other studies do in terms of the lower threshold of CDNC, for studies where humans were irrelevant (e.g. LGM)?

- *This is a very interesting question. For PMIP4, the LGM experiments should generally have the same protocols and external forcings than the CMIP6 DECK piControl simulations for 1850 (Kageyama et al. 2017), i.e. the CDNC minimum threshold seems to remain unchanged. However, aerosols and especially aerosol-cloud interactions have not yet been the focus of LGM studies, with the exception of dust (Albani et al. 2018). The study by Takemura et al. (2009) analyses the radiative effect of soil dust in the LGM, and their model further includes indirect aerosol effects. They found that "the positive radiative forcing from the indirect effect of soil dust aerosols is mainly caused by their properties to act as ice nuclei". However, they refer in their paper to the study by Takemura et al. (2005), where a minimum CDNC of 300 cm$^{-3}$ over land and 30 cm$^{-3}$ over ocean is mentioned. We therefore expect that indirect effects that affect liquid clouds (e.g. the Twomey effect) are heavily suppressed.*

- 15.16 and 19.8: I believe S8-12 contain useful information and they should not be in the supplement. Their discussion should be largely expanded as well.

- *As mentioned previously, we decided to shift the focus of this paper to the technical aspects. Therefore, we decided to leave the tables in the supplement.*

- 24.6: "5 times stronger" refers to values over land only?

- *The "5 times stronger" referred to both land and sea. Since we use the land surface temperature in the rest of the paper, we changed the sentence. Over land, the increase is 4 times stronger.*

- Page 25, first half: Why discuss so much about nitrate aerosols, which were not included in the study?

- *The omission of nitrate aerosols in the model could lead to regionally too low CCN concentrations. However, we agree with the reviewer that our discussion on this topic was too long and we shortened it accordingly.*

- 26.21: Please add "strongly" in front of "underestimated".

- *Done.*

- Section S3 mentions interannual variability. Is this in a transient or climatological sense?

- *In a climatological sense; there is no trend in fire emissions over the years.*

- Section S4, page 4, middle: Why not scale $a\_n$ with population density or crop use (or both)? Even an empirical scaling would have been better than a constant value with time, something like what was done with the vegetated area per gridbox.

- *This section of the text was not well formulated. The fire emissions for AD 1850 depend on the population density. $a\_n$ is just a scaling factor which should reflect cultural differences. We rephrased the section to make this clearer.*

- Section S4, page 4, middle: "over 1830 and 1840" means literally these years, or a climatology from 1830 to 1840, or something else?

- *It means literally these years; data is available on a 10-year resolution. We rephrased the sentence.*

**Technical corrections**

- Page 1, line 6: Please explain HYDE and KK11 in the abstract.

- *We changed the abstract and do not mention HYDE11 and KK11 anymore.*

- 6.13: Is "e.g." correct, or "i.e." was meant to be there? In either case, this sounds like a vague statement, where a solid reference should be provided for the model description.

- *The evaluation paper of the VBS model (planned to be submitted to GMD) is unfortunately still in preparation, but will be submitted soon. The model is described briefly in Mielonen et al. (2018) and Stadtler et al. (2018). We added these and other references as well as some text to describe the VBS.*

- 12.32: "Sect. S7" should had been "Sect. S6"? S7 seems irrelevant there.

- *Yes, thank you for the careful reading, we corrected it.*

- 16.7: Please add a comma after "harvest".

- *Done.*

- 24.5: What is the depth of the mixed-layer ocean?

- *50 m; we now mention this in the text.*

- 24.20: Please fix typo "smoldering".

- *Done.*

- Table S1: please change "degree" to "degrees".

- *Done.*

- Table S2 legend: Please define what is meant by "some" aerosol emissions.

- *We changed the sentence to: "Overview of natural fire aerosol, anthropogenic aerosol, and SOA precursor emissions in the different ECHAM-HAM-SALSA simulations."*

- Section S3 is very interesting and important, similar to section 2.5 in the main text, I feel it belongs to the main manuscript text.

- *Done.*

- Section S3, line 4: please define what is meant by "some" selected 30-year periods.

- *We changed the sentence to: "... from which output at a high temporal resolution was saved for a few selected 30-year periods (among them a period around AD 1 and one around AD 1835)."*

***References (not included in the paper):***

- Kageyama, M. et al.: The PMIP4 contribution to CMIP6 – Part 4: Scientific objectives and experimental design of the PMIP4-CMIP6 Last Glacial Maximum experiments and PMIP4 sensitivity experiments, Geosci. Model Dev., 10(11), 2017

- Albani, S. et al.: Aerosol-Climate Interactions During the Last Glacial Maximum, Current Climate Change Reports, 4(2), 2018

- Takemura, T. et al.: A simulation of the global distribution and radiative forcing of soil dust aerosols at the Last Glacial Maximum, Atmos. Chem. Phys., 9(9), 2009

- Takemura, T. et al.: Simulation of climate response to aerosol direct and indirect effects with aerosol transport-radiation model, J. Geophys. Res.-Atmos., 110(D2), 2005

---

## Author Comment (AC3) · 30 Aug 2019

[revised manuscript text omitted]

$$EM_{\mathrm{pasture}} = Fr_{\mathrm{pasture}} \cdot Fr_{\mathrm{pasture\_burnt}} \cdot F \cdot EF_{\mathrm{pasture}}, \tag{6}$$

where $Fr_{\mathrm{pasture}}$ is the fraction of the total gridbox covered by pasture, $Fr_{\mathrm{pasture\_burnt}}$ is the fraction of pasture that is burnt per time [$\mathrm{s^{-1}}$], $F$ stands for fuel biomass consumption [$\mathrm{kg_{dry\_matter}\,m^{-2}}$; i.e. the amount of fuel that is actually burnt], and $EF_{\mathrm{pasture}}$ is the emission factor of pasture burning [$\mathrm{kg_{aerosol}\,kg_{dry\_matter}^{-1}}$]. In accordance with the crop residue burning

**Table 3.** The values used to calculate the aerosol emissions from pasture burning (Equation 6) for the low, the intermediate, and the high scenarios. Note that the total area of pasture in the study domain ($Area_{pasture}$) is shown instead of the fraction per gridbox ($Fr_{pasture}$).

[revised manuscript text omitted]

---

## Author Comment (AC4) · 30 Aug 2019

[revised manuscript text omitted]

accordance with the crop residue burning emissions, we used the land cover reconstructions from HYDE11 for the low and the intermediate scenarios and the land cover estimates from KK11 for the high emission scenario. With this approach, the pasture area per person lies in a range between 0.58 ha and 1.0 ha for the different emission scenarios, which is similar to the values (0.56 ha and 1.05 ha, respectively) mentioned in Klein Goldewijk et al. (2011, 2017) but somewhat lower than the

5      number derived in a case study for Greece in the Roman period (1.75 ha; Weiberg et al., 2019). We considered low, intermediate, and high estimates for  $Fr_{\text{pasture\_burnt}}$ and the emission factors because these variables have large uncertainties.

**2.7.1   Fuel biomass consumption $F$**

The increase in European grassland productivity over the last decades has been small compared to crop (Smit et al., 2008). The

10      spatial variability of grassland productivity is quite large within Europe, ranging from $\approx 0.15\,\text{kg}\,\text{m}^{-2}\,\text{yr}^{-1}$ in the Mediterranean region up to $0.65\,\text{kg}\,\text{m}^{-2}\,\text{
[revised manuscript text omitted]

---

## Author Comment (AC5) · 30 Aug 2019

Attached is the new version of the supplementary material.

Please also note the supplement to this comment:
https://www.clim-past-discuss.net/cp-2019-56/cp-2019-56-AC5-supplement.pdf

———————————————————

---

## Author Comment (AC6) · 30 Aug 2019

**S1 Map of the Roman Empire**

Fig. S1: The Roman Empire at its greatest extent (AD 117) with its vassals in pink. Adapted from Tataryn (2016); licensed under the Creative Commons Attribution-Share Alike 3.0 Unported license (https://creativecommons.org/licenses/by-sa/3.0/deed.en).

**S2 Model set-upsetup**

Illustrated are the setups of the 6 simulations conducted with ECHAM-HAM-SALSA (no\_humans, LCC\_HYDE, LCC\_KK, LCC\_HYDE\_low, LCC\_HYDE\_int, and LCC\_KK\_high). Models that we used are shown in dark colours, whereas inputs to these models are shown in light colours. ECHAM-HAM-SALSA (violet) includes (among other components) the vegetation model JSBACH, a secondary organic acrosol scheme, and a sulfur cycle. Natural fire emissions were calculated with CBALONE-SPITFIRE (orange). For driving the two models, output from the Earth System Model MPI-ESM was used among others.

Tab. S1: Greenhouse gas concentrations and orbital parameters. Precession is expressed asthe longitude of the perihelion with respect to the equinox. The values are averagesover AD 50 to 150.

| Var.         | Unit          | Value   |
|--------------|---------------|---------|
| $CO_2$       | ppm           | 278     |
| $CH_4$       | ppb           | 662     |
| $N_2O$       | ppb           | 267     |
| Eccentricity | -             | 0.01742 |
| Precession   | degreedegrees | 250.8   |
| Obliquity    | degreedegrees | 23.68   |

**Tab. S2:** Overview of some-natural fire aerosol, anthropogenic aerosol, and SOA precursor 
[revised manuscript text omitted]

$$EF_{\rm ch_w} = \frac{1}{7} \cdot EF_{\rm chb} + EF_{\rm chm},\tag{1}$$

where  $EF_{chb}$  is the emission factor for charcoal burning (per kilogramme of charcoal) and  $EF_{chm}$  is the emission factor for charcoal making (per kilogramme of wood). The emission factors for  $EF_{chb}$  and  $EF_{chm}$  can be found in Tables S5 and S6, respectively.

Considering that charcoal is often cited as almost smokeless (e.g. Wood and Baldwin, 1985; Lohri et al., 2016), the measured  $EF_{\rm chb}$  of BC (median: 0.59 g kg-1; comparable to the burning of other types of biofuel) is relatively high – a discrepancy that already Bond et al. (2004) noted for the emission factors of total suspended particles. In the future, more measurements could help to better understand this inconsistency.

Inserting the estimates of  $EF_{chb}$  and  $EF_{chm}$  in the equation above results in  $EF_{chw}$  of 0.26 g kg-1wood (0.18, 0.39) for BC, 3.94 g kg-1wood (3.27, 4.80) for OC, and 0.21 g kg-1wood (0.14, 0.32) for SO2.

**S3.4 Combining different sectors**

To assess the overall emission factor, we needed to estimate how much which sector contributed to the total fuel consumption. Except for  $SO_2$ , the emission factors are similar for wood and agricultural waste. The OC emission factors for charcoal burning and production (expressed as per kilogramme wood) are larger than for wood and agricultural waste, whereas the opposite is the case for BC. Thus, different assumptions concerning the contributions from the three sectors would affect the BC to OC ratio, rather than the overall emissions of both of them.

Wood fuel (including wood for charcoal making) was the dominant fuel in the Roman Empire (Olson, 1991). However, agricultural residues such as chaff, olive pits, and dung were also used, most evidently in regions lacking in supplies of wood (e.g. Roman North Africa and Roman East; Mietz, 2016; Rowan, 2015). In developing countries in 1985, the mass contribution of agricultural waste to total biofuel combustion (excluding burning in fields) ranges from 14% in Africa to over 40% in Asia (Yevich and Logan, 2003). Based on these numbers, we assumed that 20% of the used fuel consisted of agricultural waste.

Like for developing countries (Wood and Baldwin, 1985; Yevich and Logan, 2003; Lohri et al., 2016), the use of charcoal was especially important in urban areas in ancient times (Veal, 2017). Veal (2017) assumes that in the cities "perhaps 80%" of the burnt fuel consisted of charcoal with the remainder being wood, whereas the opposite ratio occurred in rural areas. In her two extreme case scenarios for Rome, she used charcoal contributions of 80% and 20% to total fuel. Assuming a conversion factor of 7, this means that 97% and 64% of the wood fuel was used for charcoal making, respectively.

These estimates are higher than present-day estimates in countries where charcoal is proj

duced: using data from the food and agricultural organisation (FAO1), we calculated the contribution of charcoal production to total wood fuel production. We chose the year 1970, when fossil fuels were likely (even) less common in developing countries than today. A factor of 6 was used2 to convert the weight of charcoal (metric tonnes) to the volume of wood (m3) required to make the specified charcoal weight. It is unclear whether countries include the amount of wood used for charcoal making in the woodfuel statistics which they report to FAO3. To account for the fact that the data might be inconsistent, we used two methods (DIV1 and DIV2) to calculate the contribution of charcoal to wood fuel ( $frac_{DIV1}$  and  $frac_{DIV2}$ ):

$$frac_{\rm DIV1} = \frac{Wood_{\rm ch}}{Wood_{\rm con} + Wood_{\rm noncon}}$$
(2)

and

$$frac_{\rm DIV2} = \frac{Wood_{\rm ch}}{Wood_{\rm ch} + Wood_{\rm con} + Wood_{\rm noncon}},\tag{3}$$

[revised manuscript text omitted]
 0.64 kg m-2 were measured in Idaho ; values for grasslands in Australia are around 0.32 kg m-2 and 0.46 kg m-2 ; in South Africa, total aboveground fuel loads in savanna parklands range from 0.22 to  $0.55 \text{ kg m}^{-2}$ , whereas the fuel loads of standing herbaceous material range from 0.30 to 0.41 kg m-2; a total

fuel load of  $0.49 \,\mathrm{kg}\,\mathrm{m}^{-2}$  for Mediterranean grasslands in Greece was measured .

Fuel biomass consumptions for savannas are comparable:  $0.26 \text{ kg m}^{-2}$  for savanna woodlands for early dry season burns,  $0.46 \text{ kg m}^{-2}$  for savanna woodlands for mid/late dry season burns,  $0.21 \text{ kg m}^{-2}$  for savanna grasslands for early dry season burns, and  $1.0 \text{ kg m}^{-2}$  for savanna grasslands for mid/late dry season burns .

Based on these studies, we roughly estimate that the fuel biomass consumption is  $F = 0.35 \,\mathrm{kg}_{\mathrm{dry matter}} \,\mathrm{m}^{-2}$ .

**S5 Estimating**

In general, the abundance of prescribed burning depends on the accumulation of biomass: the higher the accumulation, the shorter the fire interval is. As a consequence, the fire interval depends on rainfall and grazing pressure, thus showing pronounced regional variability.

For phryganic rangelands in Greece, it is recommended to set fire every 3 to 4 years, which allows to have a good herbage production and at the same time to suppress undesirable dwarf shrub. In South Africa, Oluwole et al. (2008) found that the recovery period should be 3 years for optimum productivity in the absence of grazing. In line with this, the Burning Guidelines of South Africa do not recommend to burn pasture every year (what some farmers do), but every 2-5 years in mesic and coastal grasslands and only when it is needed in dry highveld grasslands. Smith et al. (2013) found that grass richness, evenness, and diversity was high for sites with high rainfall when frequent burning was applied in the dry season (1- to 3-year return intervals), whereas Little et al. (2015) conclude that annual burning combined with intensive grazing has a detrimental effect on plant species diversity and structure. In Australia, single fires caused a short-term reduction of yield and cover of pastures in the following year, but fast recovery occurred for most burning regimes. However, perennial grasses were reduced on the costs of annual grasses, which is why burning every 5-6 and 4-6 years for arid short grass and ribbon blue grass, respectively, are recommended. This is in agreement with the findings of Norman (1963) and Norman (1969) for native pasture on Tippera clay loam in the Katherine region. For North America, the recommended fire-return-interval of prescribed patch burning is 3 years in areas with rainfall above  $\approx 760 \,\mathrm{mm}$  per year and 4 years in drier regions.

One could argue that farmers in the past did not necessarily follow these present-day guidelines. However, traditional knowledge of prescribed burning has been lost in many European areas . Guidelines thus partially re-establish knowledge that our ancestors had. On the one hand, if burning every year reduces the productivity of many grasslands, we think that it is unlikely that ancient farmers conducted burning so often. On the other hand, a too long period without burning is also unlikely since this e.g. allows the growth of unwanted species and can have adverse effects on the ecosystem . According to the summarised literature, burning every  $\approx 3$  years seems to be a reasonable intermediate estimate. Therefore, we assumed that 30% of the pasture area is burnt per year for the intermediate scenario. For the low and the high emission scenarios, we changed this fraction by a factor of 2 and thus arrive at 15% and 60%, respectively.

**S5 Seasonal impact of anthropogenic land cover change**

**Tab. S8:** Absolute values of the reference simulation no\_human ("no") and the simulation LCC\_HYDE ("H") for each season (JJA=summer, SON=autumn, DJF=winter, and MAM=spring) and the whole year: cloud droplet number concentration burden, liquid water path, cloud cover, cloud radiative effect, precipitation, wind velocity at 10 m altitude, surface albedo over land, evaporative fraction, and turbulent flux, land surface temperature, and 2m-temperature. The values are averaged from 10° W to 50° Eand from , 20° N to 60° N. Changes Relative changes are shown in brackets (relative except for temperature) and the stars indicate significant changes (5% significance level; N = 20).

| Var.                         | Unit                   | no JJA | H JJA              | no SON | H SON               | no DJF | H DJF         | no MAM | H MAM               | no year | H year          |
|------------------------------|------------------------|--------|--------------------|--------|---------------------|--------|---------------|--------|---------------------|---------|-----------------|
| CDNC                         | $10^{9}{\rm m}^{-2}$   | 31.46  | $30.54 \ (-2.9\%)$ | 27.12  | $25.66 \ (-5.4\%)$  | 12.85  | 13.19 (2.6%)  | 22.49  | $21.92 \ (-2.5\%)$  | 23.52   | 22.87 (-2.8%)   |
| LWP                          | ${ m g~m^{-2}}$        | 54.96  | 54.89~(-0.1%)      | 59.03  | 56.36  (-4.5%)      | 39.22  | 39.10~(-0.3%) | 41.49  | 40.13~(-3.3%)       | 48.69   | 47.64~(-2.2%)   |
| CC                           | -                      | 0.32   | 0.31~(-3.1%)       | 0.50   | 0.51 (2.1%)         | 0.58   | 0.58  (1.2%)  | 0.54   | 0.54~(-0.7%)        | 0.48    | 0.48  (0.2%)    |
| CRE                          | ${ m W}~{ m m}^{-2}$   | -22.44 | -22.58 (0.6%)      | -4.23  | $-2.91 \ (-31.1\%)$ | 4.57   | 4.60  (0.5%)  | -14.43 | $-14.12 \ (-2.1\%)$ | -9.21   | -8.83 $(-4.1%)$ |
| $Wind_{10}$                  | ${\rm m~s^{-1}}$       | 4.02   | 4.05  (0.8%)       | 4.09   | 4.14 (1.1%)         | 4.63   | 4.66  (0.6%)  | 4.33   | 4.37 (1.1%)         | 4.27    | 4.30  (0.9%)*   |
| Albedo                       | -                      | 0.26   | 0.26  (0.2%)*      | 0.25   | 0.25  (-0.0%)       | 0.26   | 0.26~(-0.3%)  | 0.25   | 0.25~(-0.2%)        | 0.26    | 0.26~(-0.1%)    |
| Evap_frac                    | -                      | 0.36   | 0.36~(-1.2%)       | 0.32   | 0.33 (0.8%)         | 0.32   | 0.33 (3.2%)*  | 0.43   | 0.43  (0.7%)        | 0.36    | 0.36  (0.8%)    |
| $\mathrm{F}_{\mathrm{turb}}$ | ${\rm W}~{\rm m}^{-2}$ | 90.23  | 89.66~(-0.6%)      | 64.62  | 64.80 (0.3%)        | 52.26  | 52.73 (0.9%)  | 79.54  | 79.98  (0.5%)       | 71.78   | 71.91  (0.2%)   |

Tab. S9: The same as Table S8 but for the simulations no\_human and LCC\_KK ("KK").

| Var.                         | Unit                 | no JJA | KK JJA             | no SON | KK SON              | no DJF | KK DJF         | no MAM | KK MAM              | no year | KK year             |
|------------------------------|----------------------|--------|--------------------|--------|---------------------|--------|----------------|--------|---------------------|---------|---------------------|
| CDNC                         | $10^9 {\rm m}^{-2}$  | 31.46  | $29.84 \ (-5.2\%)$ | 27.12  | 25.51  (-5.9%)      | 12.85  | 11.73 (-8.7%)* | 22.49  | $20.36 \ (-9.5\%)*$ | 23.52   | $21.90 \ (-6.9\%)*$ |
| LWP                          | ${ m g~m^{-2}}$      | 54.96  | $53.48 \ (-2.7\%)$ | 59.03  | 55.95  (-5.2%)      | 39.22  | 36.07 (-8.1%)  | 41.49  | 37.96 (-8.5%)*      | 48.69   | 45.89 (-5.8%)*      |
| $\mathbf{C}\mathbf{C}$       | -                    | 0.32   | $0.31 \ (-1.7\%)$  | 0.50   | 0.50 (1.3%)         | 0.58   | 0.58  (0.1%)   | 0.54   | 0.53~(-1.5%)        | 0.48    | 0.48~(-0.3%)        |
| CRE                          | ${ m W}~{ m m}^{-2}$ | -22.44 | -22.20 $(-1.1%)$   | -4.23  | $-3.13 \ (-25.8\%)$ | 4.57   | 4.59  (0.4%)   | -14.43 | -13.28 $(-8.0%)*$   | -9.21   | -8.58 (-6.8%)*      |
| $Wind_{10}$                  | ${\rm m~s^{-1}}$     | 4.02   | 4.18  (3.9%)*      | 4.09   | 4.22  (3.2%)        | 4.63   | 4.74 (2.4%)    | 4.33   | 4.49 (3.7%)*        | 4.27    | 4.41  (3.3%)*       |
| Albedo                       | -                    | 0.26   | 0.26  (0.8%)*      | 0.25   | 0.25 (0.2%)         | 0.26   | 0.26  (0.0%)   | 0.25   | 0.25~(-0.5%)*       | 0.26    | 0.26  (0.1%)        |
| Evap_frac                    | -                    | 0.36   | 0.36 (1.1%)        | 0.32   | 0.33 (1.3%)         | 0.32   | 0.33 (3.9%)*   | 0.43   | 0.43  (0.6%)        | 0.36    | 0.36  (1.6%)*       |
| $\mathrm{F}_{\mathrm{turb}}$ | $W m^{-2}$           | 90.23  | $90.20 \ (-0.0\%)$ | 64.62  | 64.13  (-0.8%)      | 52.26  | 52.30 (0.1%)   | 79.54  | 80.87 (1.7%)        | 71.78   | 71.99 (0.3%)        |

**S6 Comparison of OC and SO2 emissions from anthropogenic aerosol sources and natural fire emissions in around AD 100and 
[revised manuscript text omitted]